# Effects of Simultaneous Co-Fermentation of Five Indigenous Non-*Saccharomyces* Strains with *S. cerevisiae* on Vidal Icewine Aroma Quality

**DOI:** 10.3390/foods10071452

**Published:** 2021-06-22

**Authors:** Qian Ge, Chunfeng Guo, Jing Zhang, Yue Yan, Danqing Zhao, Caihong Li, Xiangyu Sun, Tingting Ma, Tianli Yue, Yahong Yuan

**Affiliations:** 1College of Food Science and Engineering, Northwest A&F University, Yangling 712100, China; ge_qian1116@163.com (Q.G.); gcf@nwafu.edu.cn (C.G.); sunxaingyu@nwafu.edu.cn (X.S.); matingting@nwafu.edu.cn (T.M.); Yuetl@nwafu.edu.cn (T.Y.); 2Institute of Quality Standard and Testing Technology for Agro-Products of Ningxia, Yinchuan 750002, China; zhangjing2062350@163.com (J.Z.); 18895003827@163.com (Y.Y.); ZDQ_6264@163.com (D.Z.); lch.6868@163.com (C.L.); 3National Engineering Research Center of Agriculture Integration Test (Yangling), Yangling 712100, China; 4College of Food Science and Technology, Northwest University, Xi’an 710069, China

**Keywords:** Vidal icewine, non-*Saccharomyces* yeast, *S. crataegensis*, simultaneous fermentation, aromatic profile

## Abstract

In this study, Vidal grape must was fermented using commercial *Saccharomyces cerevisiae* F33 in pure culture as a control and in mixed culture with five indigenous non-*Saccharomyces* yeast strains (*Hanseniaspora uvarum* QTX22, *Saccharomycopsis crataegensis* YC30, *Pichia kluyveri* HSP14, *Metschnikowia pulcherrima* YC12, and *Rhodosporidiobolus lusitaniae* QTX15) through simultaneous fermentation in a 1:1 ratio. Simultaneous fermentation inhibited the growth of *S. cerevisiae* F33 and delayed the time to reach the maximum biomass. Compared with pure fermentation, the contents of polyphenols, acetic esters, ethyl esters, other esters, and terpenes were increased by *R. lusitaniae* QTX15, *S. crataegensis* YC30, and *P. kluyveri* HSP14 through simultaneous fermentation. *S. crataegensis* YC30 produced the highest total aroma activity and the most abundant aroma substances of all the wine samples. The odor activity values of 1 C13-norisoprenoid, 3 terpenes, 6 acetic esters, and 10 ethyl esters improved significantly, and three lactones (δ-decalactone, γ-nonalactone, and γ-decalactone) related to coconut and creamy flavor were only found in this wine. Moreover, this sample showed obvious “floral” and “fruity” note odor due to having the highest amount of ethyl ester aromatic substances and cinnamene, linalool, citronellol, β-damascenone, isoamyl ethanoate, benzylcarbinyl acetate, isobutyl acetate, etc. We suggest that simultaneous fermentation of *S. crataegensis* YC30 with *S. cerevisiae* might represent a novel strategy for the future production of Vidal icewine.

## 1. Introduction

Icewine is a kind of sweet wine that is made by delaying the harvest, allowing the grapes to hang on the vine for a certain period of time, followed by freezing, harvesting, and pressing the grapes at low temperature, and brewing [1]. Delaying harvesting increases the concentrations of sugars, acids, aroma compounds, and some non-volatile substances in the grapes, intensifying the flavor characteristics. The methods of freezing grapes, low-temperature maceration, and fermentation technology better maintain the aroma substances in the wine [2]. The current research on icewine has mainly focused on the identification of characteristic flavor substances [3] and the analysis of the dynamic changes in yeast populations during fermentation [2]. However, relatively few results have been reported on the influence of mixed fermentation with non-*Saccharomyces* and *Saccharomyces* yeasts on the enhancement in icewine flavor quality.

Wine fermentation is a complex biochemical process in which yeasts play a critical role by converting sugar into ethanol, carbon dioxide, and thousands of other secondary metabolites [4]. Scientific research showed that the quality of wine is highly dependent on the metabolic activities and fermentation behavior of different yeasts, which significantly contribute to the chemical composition, and sensory and flavor characteristics of wine [5]. To date, *S. cerevisiae* is the most widely used strain for wine production in the industry [6], mainly due to its ability to control the risk of deterioration and its good fermentation power [7], but its use is associated with problems regarding the singularity and homogeneity of wine flavor characteristics [8]. The fermentation strategy of pure starter cultures has drastically reduced the diversity of the yeast species and clones involved in winemaking, resulting in a uniformly plain flavor of the product [7]. As such, using mixed *S. cerevisiae* and non-*Saccharomyces* yeast fermentation has become a strategy pursued by winemakers, especially using some indigenous yeasts that have strong adaptability and representativeness, to obtain a unique style of wine that has representative, diverse, and complex aroma characteristics, thereby improving and enhancing wine flavor quality [9].

Non-*Saccharomyces* yeasts have become a choice to improve wine quality [10]. Numerous studies have demonstrated that these yeast strains can be used to achieve specific objectives such as producing some expected secondary metabolites, lowering the ethanol content, preventing the growth of some undesirable strains, and increasing the production of specific enzymes, but the disadvantage is their insufficient fermentation power [11]. Mixed fermentation of non-*Saccharomyces* and *S. cerevisiae* yeasts can not only improve the diversity and complexity of wine aroma, but also compensate for the lack of fermentation power of non-*Saccharomyces* yeasts, which is an effective method to improve wine aroma quality [12]. Li et al. [9] found that highly antagonistic *S. cerevisiae* co-inoculation with *P. fermentans* significantly improved the production of glycosidase activity and wine varietal odorants. Morales et al. [8] confirmed that *L. thermotolerans* co-inoculated with *S. cerevisiae* at the ratios of 50:1 and 20:1 enhances wine flavor quality and aromatic complexity. *S. cerevisiae* sequential inoculation fermentation with *Metschnikowia pulcherrima, Torulaspora delbrueckii and Zygosaccharomyces bailii* under 0.025 VVM aeration conditions reduced the ethanol concentration of chardonnay wine by 1.6%, 0.9%, and 1.0% (*v*/*v*), and the chemical volatile profiles of the wine were acceptable [6]. Using *Starmerella bacillaris* mixed with *S. cerevisiae* produced significantly lower levels of acetic acid, ethanol, and ethyl acetate, and higher amounts of glycerol, higher alcohols, and esters [13]. *P. kluyveri* improves wine quality parameters such as thiol, fruity ester, and terpene concentrations, mainly in sequential fermentation [14]. Higher concentration of medium-chain fatty acids was associated with the higher biomass suppression of *H. uvarum* in co-inoculation with killer *S. cerevisiae*, which resulted in the increased formation of fruity esters, but effectively restricted the production of ethyl acetate [15]. *H. uvarum*, in mixed fermentation with commercial SC F5, increased the medium-chain fatty acid ethyl ester content in both the synthetic media and grape must of cabernet Gernischt grapes [16]. *M. pulcherrima* has also been shown to increase wine flavour and aroma in Debina wines [17,18], and/or enhance positive sensory attributes, such as “citrus/grape fruit”, “pear” and “flowery” in Riesling [19] and in base wine for sparkling wine production [20].

Therefore, in this study, using the Vidal icewine as an example, we evaluated the influence of the mixed fermentation of non-*Saccharomyces* and *Saccharomyces* yeasts on the flavor characteristics of wine, explored the potential non-*Saccharomyces* yeasts that can improve Vidal icewine flavor quality, and then analyzed the potential association between characteristic aroma substances and their contribution to wine aroma. Our findings provide a new strategy for the industrial production of icewine.

## 2. Materials and Methods

### 2.1. Yeast Strains

The fermentations were performed with one commercial *S. cerevisiae* Actiflore^®^ F33 strain (Laffort) as control and five representative indigenous non-*Saccharomyces* yeast strains isolated from Ningxia province in China: *Hanseniaspora uvarum* QTX22 (Hu, MT505668), *Saccharomycopsis crataegensis* YC30 (Sc, MT505672), *Pichia kluyveri* HSP14 (Pk, MT505679), *Metschnikowia pulcherrima* YC12 (Mp, MT505675) and *Rhodosporidiobolus lusitaniae* QTX15 (Rl, MT505670). *S. crataegensis* YC30 grew on the WL medium as with a white flocculent convex colony morphology (Figure 1).

### 2.2. Reagents and Standards

The organic acid standards (L-tartaric acid, lactic acid, malic acid, acetic acid, etc.), as well as polyphenol standards (protocatechuic acid, p-hydroxybenzonic acid, chlorogenic acid, vanillic acid, (-)-epigallocatechin gallate, epicatechin, vanillin, p-coumaric acid, (-)-epicatechin gallate, isoferulic acid, etc.), were purchased from Shanghai Yuanye Bio-Technology Co., Ltd. (Shanghai, China). Water was purified using a Milli-Q system from Millipore (Bedford, MA, USA).

Isoamyl ethanoate, benzylcarbinyl acetate, isobutyl acetate, butyl acetate, propyl acetate, hexyl acetate β-damascenone, naphthalene, cinnamene, linalool, citronellol, d-limonene, α-terpineol, nerol oxide, ethyl dodecylate, ethyl heptoate, ethyl caproate, ethyl butanoate, ethyl caprylate, ethyl propanoate, ethyl isobutyrate, ethyl n-valerate, etc., were provided by Sigma-Aldrich (Shanghai, China). The internal standard of 4-methyl-1-pentanol was obtained from Tokyo Chemical Industry Co. Ltd. (Tokyo, Japan). The calculation of retention indices (RIs) was used with n-alkanes (C8-C40) purchased from Supelco (Bellefonte, PA, USA).

### 2.3. Grape Juice

*Vitis vinifera L. cv.* Vidal grape was harvested in December 2017 with temperature at −12 °C and obtained from the Chateau barges vineyard (Yinchuan, Ningxia Province, China, 106.02° E, 38.24° N), with a sugar content of 344.0 g/dm^3,^ acidity of 4.2 g/dm^3^ (as tartaric acid) and pH 3.58. The total sugar content was analyzed by the reduction method using Fehling’s reagent [21]. Total acid content was determined by titration of samples with 0.1 mol/L NaOH (tartaric acid equivalent) and ethanol content was assayed using the gas chromatographic method (GB/T 15038-2006, 2006) [22]. The agronomic practices of the region were applied to manage the vineyard and the vines were planted in 2013 with a vine row spacing of 1.0 × 2.0 m. About 80 kg of grapes were destemmed, crushed, and treated with sulfur dioxide (50 mg/L K_2_S_2_O_5_) and 20 mg/L pectinase (≥500 U/mg) purchased from Laffort Co. (Bordeaux, France), and macerated at 4 °C for 72 h to inhibit bacterial growth and increase the juice yield. The obtained grape juice was stored at −20 °C until use.

### 2.4. Fermentation Strategies

For fermentation, we followed a microfermentation method according to Wei et al. [21], with modifications. The yeast strains were stored at −80 °C with 20% sterile glycerol before use. The cryogenically preserved yeasts were propagated by two successive transfers in sterile yeast extract peptone dextrose (YPD) medium (yeast extract, peptone and glucose were 1%, 2% and 2% (*w*/*v*), respectively) for 24 h each time. Subsequently, the yeast strains were pre-cultured in pasteurized grape juice (100 °C, 10 min) for 12 h (28 °C, 2.5 g). After the yeast strains were activated to viable counts of approximately 10^6^ CFU/mL, they were inoculated into 350 mL of sterile grape juice in 500-mL Erlenmeyer flasks. The flasks were fitted with sterile glass air locks that contain sulfuric acid to allow the CO_2_ produced during the fermentation to escape to avoid microbial contamination [23]. Fermentation trials were performed using commercial *S. cerevisiae* F33 in pure culture as a control (pure_F33), and simultaneous co-fermentation with *H. uvarum* QTX22 (F33_Hu), *S. crataegensis* YC30, *P. kluyveri* HSP14 (F33_Pk), *M. pulcherrima* YC12 (F33_Mp), and *R. lusitaniae* QTX15 (F33_Rl) in a ratio of 1:1, separately. The quantity of CO_2_ released was monitored every 24 h by weighing the bottles during fermentation. Fermentations were controlled at 18 °C and continued until no more weight loss was quantified for three consecutive days, regardless of the residual sugar level. This was performed in triplicate: three flasks were inoculated with the same culture at the same time. The grape juice uninoculated with yeast strains was used as a control and incubated alongside the fermentations. One milliliter of fermenting grape juice was collected periodically (Day 0, 1, 2, 3, 4, 5, 7, 9, 12 and 13) and diluted to a suitable concentration, and then 100 μL of these cultures were plated on WL nutrient agar to facilitate yeast population counts. The colonies of *S. cerevisiae* F33 were smooth and creamy white, whereas those of *H. uvarum* QTX22 and *R. lusitaniae* QTX15 were smooth and green, those of *S. crataegensis* YC30 and *P. kluyveri* HSP14 were white with flocculent folds, and that of *M. pulcherrima* YC12 was milky white, and could be easily discriminated from *S. cerevisiae* F33 in mixed cultures on WL media. Fermentation rate was calculated as the loss of CO_2_ (g/L) within 24 h during the fermentation [2]. After fermentation, all wine samples (350 mL) were centrifuged for 8 min (4000× *g*, 4 °C), and the cell-free supernatants were stored at −20 °C and used within 6 months for analysis.

### 2.5. Assay for Organic Acids and Polyphenol Compounds

The organic acids and polyphenols samples were analyzed following the method described by Ye et al. [24], with modifications. The organic acids and polyphenolic compounds were determined by high-performance liquid chromatography (HPLC) on an LC-15C HPLC system equipped with a photodiode array detection and a SIL-10AF automatic sampler (Shimadzu, Kyoto, Japan). For the analysis of organic acids, wine samples were diluted with ultrapure water to an appropriate concentration and filtered through a 0.25 μm Waters membrane filter. The target compounds were separated on a Waters XSelect^®^ HSS T3 reversed-phase column (250 × 4.6 mm, particle size of 5.0 μm) at a total flow rate of 1 mL/min with gradient elution. Mobile phases A and B were methanol and 0.01 mol/L ammonium phosphate aqueous solution, respectively. The mobile phase A increased linearly within 12 min from 0% to 2% and then held for 13 min. For the analyses of polyphenolic compounds, 50 mL samples were adjusted to pH 7.0 and pH 2.0 with 1 mol/L NaOH and HCl, respectively, and were extracted three times with 100 mL ethyl acetate. The organic phase was combined and evaporated to dryness on a vacuum rotary evaporator at 35 °C, and then dissolved in 25 mL of methanol. The resultant solution was filtered through a 0.25 μm Waters membrane. The mobile phase A was 2% acetic acid in water (*v*/*v*), and the mobile phase B was 0.5% acetic acid in water and acetonitrile (50:50, *v*/*v*). The gradient elution was 10–55% solvent B (50 min), 55–100% B (10 min), and 100–10% B (5 min), with a flow rate of 0.8 mL/min on a Waters xTerra MS C18 reverse-phase column (250 × 4.6 mm, particle size of 5.0 μm) at 40 °C, with a 75 min total run time. The absorbance signal was read at 280 nm for dihydrochalcones and flavan-3-ol, 320 nm for hydroxycinnamic acid, and 360 nm for flavonols. Both organic acids and polyphenolic compounds quantification methods were performed using standards with the external standard and repeated three times.

### 2.6. Quantification of Volatile Compounds

Volatile compounds were analyzed by headspace solid-phase microextraction combined with gas chromatography-mass spectrometry (HS-SPME-GC-MS) on a GC-MS TQ8050 NX system (Shimadzu) equipped with an InertCap WAX chromatographic column (30 m × 0.25 mm × 0.25 μm, GL Sciences Inc., Tokyo, Japan), as described previously [25]. We placed 5 mL wine samples were placed in 20 mL headspace bottles containing 1.5 g NaCl, and 4-methyl-1-pentanol internal standard solution was added to each sample. SPME fiber (50/30 μm, DVB/CAR/PDMS, Supelco, Inc., Bellefonte, PA, USA) was inserted into the headspace bottles containing the sample solution, equilibrated in a 40 °C water bath with stirring for 15 min, extracted for 35 min, and then desorbed in the GC injector at 250 °C for 3 min. Mass spectra were acquired over m/z 50–450 in EI mode at 70 eV. The retention index (RI) based on the mixture of n-alkanes (C8-C40), retention time, mass spectra, and 85% similarity, per the NIST 14 library, were used to tentatively qualitatively analyze the compounds. Where possible, the identification of compounds was confirmed by comparing an external standard method with authentic standards. Each experiment was repeated three times with three replicates in each experiment.

The odor activity value (OAV), defined as the ratio between the concentration of the individual chemical compound and its sensory detection threshold in the literature, was calculated for all the identified volatiles to evaluate the contribution of volatiles to wine aroma [26].

### 2.7. Sensory Analysis

For our sensory analysis method, we followed a method described in the literature with modifications [27]. About 18 sensory panel students of 9 women and 9 men were trained using a 54 aroma kit (Le Nez du Vin^®^, Jean Lenoir, Provence, France), according to the wine industry, for 45 days. During the training, each aroma type was identified by the panel every six days until the panel identification deviation for each aroma item was less than 5%. Wine samples were placed in a clean black glass in a random order at a 25 °C room temperature and identified in duplicate. The wine aroma profile was described by each sensory panelist according to the terms of Le Nez du Vin aroma kit. A five-point scale was used for wine aroma profile scores: 0, no perception; 1, very weak; 2, weak; 3, medium; 4, strong; and 5, very strong. The computational formula was as follows:$$MF\%=\sqrt{F\left(\%\right)I\left(\%\right)}$$
where *F*% is the average detection frequency of the terms described in an aroma group by the panel, and *I*% is the average intensity of the described terms in the group expressed as the percentage of maximum intensity.

Triangular tests were performed to determine whether there was a perceptible difference between the two samples. During the tasting, two samples were poured into three glasses at a time. Two of them contained the same sample. Sensory panel students were asked to identify which glass was different from the other two samples. The wine samples were presented in a random order, including all possible combinations. The results of all of the triangular tests were statistically analyzed according to the tables reported in the literature [28] and based on the binomial law corresponding to the distribution of answers in this type of test.

### 2.8. Statistical Analysis

The correlation analysis between sensory analysis and aroma compounds was performed using PLSR via Unscrambler 9.7 (CAMO ASA, Trondheim, Norway). One-way analysis of variance (ANOVA) and Duncan’s multiple-range test (*p* < 0.05), column graphs, PCA, and pheatmaps were implemented using R version 3.6.1.

## 3. Results and Discussion

### 3.1. Fermentation Performance of Yeasts

Parameters including fermentation rate, along with yeasts biomass, cell numbers, etc., were found to have a critical influence on the final flavor quality of wine during fermentation [29]. Figure 2A shows the growth kinetic characteristics of pure culture of *S. cerevisiae* F33 and the mixed fermentation with five non-*Saccharomyces* yeasts. The results showed that the maximum CO_2_ production rate in the pure culture was reached on day 3 (98.2 g/L); when with non-*Saccharomyces* yeast strains, the maximum was reached on day 4 (72.8–83.7 g/L). The final CO_2_ release in pure fermentation was almost perfectly aligned with simultaneous fermentation, indicating that the fermentation of Vidal icewine fermented well under the two fermentation strategies.

The growth kinetics of *S. cerevisiae* F33 in pure and mixed fermentation with five non-*Saccharomyces* yeasts are shown in Figure 2B–G. All yeast strains propagated sharply within 1 day after inoculation; pure *S. cerevisiae* F33 reached a maximum on day 3 (8.25 log CFU/mL), and then continually decreased until it stabilized on day 7 (7.48 log CFU/mL). However, the simultaneous fermentation of five non-*Saccharomyces* yeast strains delayed the time for *S. cerevisiae* F33 to reach the maximum biomass. *S. cerevisiae* F33 reached the maximum biomass on day 3 when mixed with *S. crataegensis* YC30 (7.99 log CFU/mL) and *R. lusitaniae* QTX15 (7.99, 8.10 log CFU/mL), on day 4 when mixed with *H. uvarum* QTX22 (7.98 log CFU/mL) and *M. pulcherrima* YC12 (8.14 log CFU/mL), and on day 5 when mixed with *P. kluyveri* HSP14 (7.93 log CFU/mL). The mixed fermentation strategy inhibited and delayed the growth and development of *S. cerevisiae* F33 to different degrees, which is consistent with a previous report [21]. The wine fermented with *S. cerevisiae* F33 and *R. lusitaniae* QTX15 could not be detected after day 7, whereas *S. cerevisiae* F33 mix-fermented with other non-*Saccharomyces* yeast strains could not be detected after day 5 under this competitive relationship. Binati et al. [30] reported that the biomass of non-*Saccharomyces* yeast strains decreased sharply after inoculation with *S. cerevisiae*, especially *Metschnikowia* spp. Non-*Saccharomyces* yeast strains could not be detected after day 3 during the fermentation of pinot grigio wines, which may related to the antimicrobial peptide secreted by *S. cerevisiae* and the contact between cells [31]. In addition, although non-*Saccharomyces* yeast strains reached the maximum biomass (7.33–7.53 log CFU/mL) on day 3, the maximum biomass was all lower than that of *S. cerevisiae* F33 and inhibited the biomass of *S. cerevisiae* F33, especially *P. kluyveri* HSP14 and *H. uvarum* QTX22, suggesting that weak competition occurs between *S. cerevisiae* and non-*Saccharomyces* yeast strains. Canonico et al. reported that the mixed fermentation of *Torulaspora delbrueckii* and *S. cerevisiae* in a ratio of 1:1 can inhibit the biomass of *S. cerevisiae* and control the entire fermentation process, which plays a dominant role when increasing the inoculation amount. Hu et al. (2018b) [32] reported that *H. uvarum* has a weak competitive relationship with *S. cerevisiae*, and controlling the amount of inoculation can indirectly affect the production of wine aroma. In our study, we found that the biomass of and CO_2_ released by yeast strains were inhibited and delayed by the addition of non-*Saccharomyces* yeast strains; though timely and reliable completion of fermentation is of primary importance in the wine industry [33], low-speed fermentation can be considered a benefit for the production and retention of aroma substances in wine and a reduction in the demand for the energy requirement of yeast, which may improve the wine aroma quality [34].

### 3.2. Effect of Fermentation Strategies on Physicochemical Characteristics in Vidal Icewine

The physicochemical characteristics of grape juice and Vidal icewine are shown in Table 1 and Figure 3. The ANOVA revealed significant (*p* < 0.0001) effects for all the wine samples in terms of physicochemical data and organic acid concentration, except for pH and total acid. Compared with grape juice, the contents of total sugar in wine decreased by 79.09–90.59% and ethanol increased by 7.35–10.12%, indicating that the fermentation progressed smoothly and thoroughly. Compared with the fermentation in pure culture, simultaneous fermentation reduced the ethanol concentration by 14.42–27.27%, which might be related to the weak competition among the yeast strains [32].

The type and composition of organic acids significantly affect the sensory and chemical properties of wine, including pH, total acid, microbiological stability, etc. [35]. Some organic acids can produce more complex odors through esterification, which enhances the fruit notes of the wine [36]. Therefore, it was necessary to evaluate the effects of different fermentation strategies on the organic acids of Vidal icewine. The main organic acids in grape juice were tartaric, malic, and citric acid, accounting for 98.6% of all the individual organic acids, which is similar to the values in a previous report [37]. Pyruvate, oxalic, malonic, acetic, and lactic acid were not detected in grape juice, which are all produced by yeast metabolism during fermentation; malic, citric, and tartaric acid decreased during fermentation [38]. In the wine samples, the highest concentrations were tartaric and malic acids, followed by succinic, lactic, and acetic acids. Lactic acid, providing roundness and balanced acidity to the taste of the wine, is mainly affected by different strains [39]. *Lachancea thermotolerans* can produce more lactic acid to modify the wine’s acidity [40]. Acetic acid is a fermentation coproduct with a bitter and sour taste, which is related to the decrease in the malic acid concentration [37], and its concentration should be controlled within 0.9 g/L [41]. Rantsiou et al. [42] reported that the concentration of acetic acid in sweet wine fermented by *C. zemplinina* and *S. cerevisiae* can be reduced to 0.30 g/L due to the osmotic stress response of the yeast. In our study, simultaneous fermentation produced a higher concentration of malic acid than pure fermentation, but it had little effect on citric, oxalic, and fumaric acids. Simultaneous fermentation of *S. cerevisiae* F33 and the Pk strain produced the highest levels of pyruvate, malic, succinic, and acetic acids (0.564 g/L), whereas the lowest concentrations of acetic, succinic, and pyruvate acids were produced by Hu, Sc, and Mp strains. *Candida zemplinina* can increase malic acid content [43], whereas *Lachancea thermotolerans* can limit the production of malic acid [23,39]. In addition, *Pichia kluyveri* can increase oxalic, lactic, and succinic acids contents and malic acid degradation in wine [44,45], modifying the acidity of wine using various yeast strains.

### 3.3. Effect of Fermentation Strategies on Polyphenols in Vidal Icewine

Polyphenol, a critical quality parameter of the secondary metabolites of the grapes and wine, is not only related to wine color, astringency, bitterness, and other flavor quality parameters, but also has good biological and antioxidant properties [46]. Table 2 and Figure 4 describe the polyphenol contents of Vidal icewine produced using different fermentation strategies. Protocatechuic acid, rutin, and quercetin were the main phenolic substances in grape juice, whereas (-)-epigallocatechin gallate, (-)-epicatechin gallate, and cynaroside were not detected. The total polyphenol concentrations in mixed fermentation increased by approximately 4.70–7.69 mg/L, whereas that in the wine produced with *S. cerevisiae* F33 through pure fermentation increased by 1.91 mg/L. The main phenolic substances in wine samples produced by mixed fermentation were caffeic acid, (-)-epicatechin gallate, and p-coumaric acid, followed by protocatechuic acid, cynaroside, trans-cinnamic acid, etc. Compared with *S. cerevisiae* F33 produced by pure culturing, mixed fermentation of F33 and *H. uvarum* QTX22 significantly increased the contents of caffeic acid, chlorogenic acid, apigenin, p-hydroxybenzonic acid, etc.; *S. crataegensis* YC30 significantly increased myricetin, chlorogenic acid, quercetin, vanillic acid, p-hydroxybenzonic acid, etc., contents; *M. pulcherrima* YC12 significantly increased myricetin, vanillic acid, caffeic acid, chlorogenic acid, etc., contents. Medina et al. [34] found that the mixed fermentation of commercial *S. cerevisiae* with *Hanseniaspora vinea* increases the phenolic concentrations of chardonnay wine and improves some sensory characteristics of the wine, which is consistent with our experimental conclusions. However, Hranilovic et al. [33] reported that the mixed fermentation of *S. cerevisiae* and *M. pulcherrima* led to a decrease in the content of flavan-3-ols and anthocyanins compared with *S. cerevisiae* in pure in Syrah wines, which may be related to the raw material of grape juice.

### 3.4. Effect of Fermentation Strategies on Volatile Aroma Substances in Vidal Icewine

A total of 118 aroma compounds were detected by HS-SPME-GC-MS (Table 3 and Figure 5A,C). There were 36 common aroma substances to grape juice and wine samples under the six fermentation strategies, among which only simultaneous fermentation of *S. cerevisiae* F33 and *S. crataegensis* YC30 produced eight unique aroma substances including ethyl linoleate, δ-decalactone, γ-decalactone, γ-caprolactone, isobutyric acid, propyl propionate, isoamyl butylate, and valeric acid (Figure 5B). The main aroma substances in grape juice were terpenes, acids, and aldehydes, accounting for 74.9%; those in the wine samples were mainly acetate esters, ethyl esters, alcohols, and acids, accounting for 84.8%, which is similar to previously reported values [12]. The total content of aroma substances in wine was 24.3 times that in grape juice, which is mainly attributed to the glucosidase, protease, and pectinase secreted by yeasts through sugar and amino acid metabolism during fermentation, so that the varietal aroma substances mainly exist in grape in the form of odorless binding glycosides hydrolyzed to free volatile aroma substances, which improve the aroma quality of wine [47]. Additionally, non-*Saccharomyces* yeast strains significantly increased the total content of volatile aroma substances and the contents of acetate esters, ethyl esters, other esters, and terpenes in Vidal icewine, especially the co-culture of *S. crataegensis* YC30 and *H. uvarum* QTX22.

#### 3.4.1. Varietal and Aroma Substances

A total of 17 varieties of aroma substances were detected, including 1 C13-norisoprenoid (β-damascenone), 14 terpenes, and 2 volatile phenols (Table 3, Figure 5A,C) through the qualitative and quantitative analysis of the aroma substances of Vidal icewine. Compared with grape juice, C13-norisoprenoid (β-damascenone) and terpenes were significantly increased during the fermentation, in agreement with a previous study [9]. Terpenes and C13-norisoprenoid (β-damascenone) aroma substances are a highly important category of aroma compounds in wine that provide essential contributions to wine’s floral and fruity aroma notes [48]. Compared with pure fermentation, mixed fermentation of *S. cerevisiae* F33 with *S. crataegensis* YC30 and *P. kluyveri* HSP14 significantly increased the content of terpenes substances, which may be related to a higher β-glucosidase activity produced by *P. kluyveri* [49,50]; mixed fermentation with *H. uvarum* QTX22, *S. crataegensis* YC30, and *M. pulcherrima* YC12 significantly increased C13-norisoprenoids substances (β-damascenone); mixed fermentation with *H. uvarum* QTX22 significantly improved the content of volatile phenolic compounds by about 29.8%. Hu et al. (2018b) [32] reported that mixed fermentation of *H. uvaum* and *S. cerevisiae* increased the content of terpenes in cabernet sauvignon wine, but it also increased the volatile phenols by 53%. Although research reports are scarce on the effects of non-*Saccharomyces* yeast strains on volatile phenols, their improvement may lead to unpleasant odors such as a medicinal taste of the wine [51], which should be a focus in future production. With simultaneous fermentation, *S. crataegensis* YC30 significantly increased the concentration of β-damascenone, cinnamene, linalool, citronellol, α-terpineol, and nerol oxide contents; *H. uvarum* QTX22 significantly increased β-damascenone, cinnamene, α-terpineol, and 2,4-di-t-butylphenol contents; and *P. kluyveri* HSP14 significantly increased the contents of linalool, p-xylene, and nerol oxide. β-damascenone, cinnamene, linalool, citronellol, and nerol oxide are mainly related to floral and fruit flavors [52], which contribute to enhancing the sensory characteristics, the floral and fruity aroma notes, of wine. Beckner et al. [53] confirmed that simultaneous fermentation of *S. cerevisiae* and non-*Saccharomyces* yeast strains can increase the contents of linalool, geraniol, nerol, and ocimene in sauvignon blanc wine, and the increase in these terpenes is mainly related to some active enzymes such as β-glycosidase.

#### 3.4.2. Fermentative Aroma Compounds

##### Esters

Esters are one of the main fermentation products second only to alcohols in wine, and include acetate esters, ethyl esters, and other esters [21]. A total of 50 esters were detected, including 14 acetate esters, 20 ethyl esters, and 16 other esters (Table 3, Figure 5A,C). Among acetate ester aromatic compounds, only four acetate aromatic compounds were detected in grape juice, and most of them were produced during fermentation by the yeast strains [32]. In the wine samples, the main aroma substances were propyl acetate, benzylcarbinyl acetate, butyl acetate, isobutyl acetate, and ethyl acetate, and all five non-*Saccharomyces* yeast strains significantly increased the acetate ester aromatic compound contents, especially *P. kluyveri* HSP14, *H. uvarum* QTX22, and *S. crataegensis* YC30, all of which increased the contents of butyl acetate, propyl acetate, and hexyl acetate. Additionally, both *S. crataegensis* YC30 and *P. kluyveri* HSP14 increased the contents of methyl phenylacetate and furfuryl acetate, *P. kluyveri* HSP14 and *H. uvarum* QTX22 increased the content of 9-decenyl acetate, and *S. crataegensis* YC30 and *H. uvarum* QTX22 significantly increased the contents of benzylcarbinyl acetate, isobutyl acetate, and isoamyl ethanoate. In particular, among the five non-*Saccharomyces* yeast strains, only *P. kluyveri* HSP14 significantly increased the contents of amyl acetate and isooctyl acetate, and only *H. uvarum* QTX22 increased the contents of ethyl acetate, methyl acetate, and heptyl acetate. *M. pulcherrima* YC12 and *R. lusitaniae* QTX15 showed a weaker ability to enhance the acetate esters aroma substances content of wine. The ability of *H. uvarum* QTX22 to increase the content of acetate ester substances in wine has been reported [32,54]. Wei et al. [44] also confirmed that *P. kluyveri* HSP14 can significantly increase the aroma substances of cider acetate. In this study, we found that in addition to *H. uvarum* QTX22 and *P. kluyveri* HSP14, *S. crataegensis* YC30 also has a good ability to produce acetate substances.

Among ethyl ester substances, there were six ethyl ester compounds; about 0.155 mg/L was detected in grape juice. We detected 18 ethyl ester compounds, about 12.57–17.58 mg/L, in wine samples, and the main aroma substances were ethyl caproate, ethyl dodecylate, ethyl 4e−decenoate, ethyl propanoate, an ethyl 7−octenoate. *S. cerevisiae* F33 mixed-fermented with *H. uvarum* QTX22, *S. crataegensis* YC30, and *P. kluyveri* HSP14 significantly increased ethyl ester aroma substances. Among them, *S. crataegensis* YC30 significantly increased the concentration of 17 aroma substances including ethyl dodecylate, ethyl heptoate, ethyl caproate, ethyl butanoate, ethyl myristate, ethyl nonanoate, ethyl palmitate, ethyl lactate, and ethyl dihydrocinnamate; *H. uvarum* QTX22 significantly increased 12 kinds of aroma substance concentration, including ethyl dodecylate, ethyl heptoate, and ethyl caproate, ethyl butanoate, ethyl myristate, ethyl nonanoate, etc.; and *P. kluyveri* HSP14 significantly increased the concentration of six kinds of aroma substances: ethyl propanoate, ethyl isobutyrate, ethyl dihydrocinnamate, ethyl myristate, ethyl dodecylate, and ethyl caproate. The reported results of ethyl ester aroma substances from mixed fermentation vary; Ye et al. [24] reported that mixed fermentation of *S. cerevisiae* and *Wickerhamomyces anomalus* increased the content of ethyl ester aroma substances, whereas Renault et al. [28] reported that mixed fermentation with *Torulaspora delbrueckii* had little effect on the content of ethyl ester aroma substances. Varela et al. [55] suggested that mixed fermentation with *M. pulcherrima* resulted in a 16% reduction in ethyl ester aroma substances; Hu et al. [56] found that mixed fermentation with *H. uvarum* can increase the content of ethyl esters, especially the medium-chain fatty acid ethyl ester content. In our study, only *H. uvarum* QTX22, *S. crataegensis* YC30, and *P. kluyveri* HSP14 strains mixed with *S. cerevisiae* F33 increased the content of ethyl ester aroma substances in Vidal icewine, whereas *R. lusitaniae* QTX15 and *M. pulcherrima* YC12 had a non-significant main effect on the increase in ethyl ester aroma substances.

For other esters, *H. uvarum* QTX22, *S. crataegensis* YC30, and *P. kluyveri* HSP14 produced a significant effect. *S. crataegensis* YC30 significantly enhanced 14 other ester aroma substances, of which 3 lactone aroma substances, including δ-decalactone, γ-decalactone, and γ-caprolactone, were not detected in other grape juice and wine samples nor were isobutyric acid, propyl propionate, and isoamyl butylate. In addition, all five non-*Saccharomyces* yeast strains significantly increased the concentration of γ-nonalactone and isoamyl decanoate; *H. uvarum* QTX22, *S. crataegensis* YC30, and *P. kluyveri* HSP14 significantly increased methyl caprate and isobutyl octanoate; and *S. crataegensis* YC30, *H. uvarum* QTX22, and *R. lusitaniae* QTX15 significantly increased the concentration of vinyl acetate. Some lactones, such as δ-decalactone, γ-nonalactone, γ-decalactone, and γ-caprolactone, are mainly related to coconut, creamy, milky, fruity, and sweet odor characteristics [57], and are produced by the corresponding hydroxy acids [58], so they further improve the complexity of the aroma quality of wine.

##### Alcohols

A total of 24 alcohols were detected in the grape juice and wine samples (Table 3, Figure 5A,C). Although the highest alcohol content was detected in grape juice, it significantly increased during fermentation, and the alcohol contents in the wine samples were approximately 30–46 times higher than in grape juice. Wei et al. [21] reported that the alcohol content of cider was 16–30 times higher than that of apple juice. A large amount of phenylethyl alcohol was detected in cider, but not in apple juice, which is consistent with our findings. The main alcohol compounds found in our wine samples were isobutyl alcohol, (z)-2,3-butanediol, 1-heptanol, 1-hexanol, phenethyl alcohol, etc. Compared with pure fermentation, only mixed fermentation with *H. uvarum* QTX22 increased the total alcohol aroma compounds content, mainly including those of 13 aroma compounds: 1-pentanol, 1-dodecanol, 1-butanol, isobutyl alcohol, benzyl alcohol, hotrienol, hexanol, penten-3-ol, 2-heptanol, (z)-3-hexen-1-ol, etc. Additionally, *S. crataegensis* YC30 significantly increased the contents of 11 alcohol aroma compounds, such as 1-dodecanol, benzyl alcohol, hexanol, penten-3-ol, 2-heptanol, (z)-2,3-butanediol, (z)-3-hexen-1-o, etc.; *P. kluyveri* HSP14 and *R. lusitaniae* QTX15 significantly increased the contents of six and seven alcohol aroma compounds, respectively. Benzyl alcohol, hexanol, penten-3-ol, 2-heptanol, and (z)-3-hexen-1-ol are mainly responsible for fruity, floral, rose and other aroma characteristics of wine, which are desired in wine aroma [59]. The complexity and the fruity, floral, and rose aroma notes were further enhanced by the mixed fermentation of *H. uvarum* QTX22 and *S. crataegensis* YC30 with *S. cerevisiae* F33. The higher alcohol content is related to the enzymatic activity of alcohol acetyltransferase, which is esterified under the action of acetyltransferase [41]. The alcohol aroma compounds content produced by the fermentation of *S. cerevisiae* F33 mixed with *H. uvarum* QTX22 was the highest. Among them, isobutyl alcohol is mainly produced from the catabolism of valine [60].

##### Acids

Acids are mainly produced during fermentation, and the concentrations of acids in the wine were 12–24 times higher than in grape juice (Table 3, Figure 5A,C). A total of 13 acids were detected, mainly including acetic acid, octanoic acid, n-decanoic acid, 9-decenoic acid, heptanoic acid, etc. Compared with pure fermentation, only mixed fermentation with *H. uvarum* QTX22 significantly increased the content of acids, mainly including octanoic acid, n-decanoic acid, butanoic acid, 9-decenoic acid, 2-methyl-butanoic acid, heptanoic acid, and 2-oxooctanoic acid. In addition, mixed fermentation with *S. crataegensis* YC30 significantly increased the concentrations of acetic acid, 2-methyl-butanoic acid, heptanoic acid, 2-methyl-propanoic acid, propanoic acid, and valeric acid. *P. kluyveri* HSP14 significantly increased the concentrations of 2-methyl-butanoic acid, heptanoic acid, 2-methyl-propanoic acid, hexanoic acid, and nonanoic acid. Both *M. pulcherrima* YC12 and *R. lusitaniae* QTX15 significantly increased the concentrations of heptanoic acid, hexanoic acid, and nonanoic acid; and *R. lusitaniae* QTX15 also significantly increased the concentration of acetic acid. A high concentration of volatile acetic acid can produce a cat urine taste in wine, which deleteriously affects the aroma quality [2]. Canonico et al. [61] reported that the concentration of acetic acid in beer produced by mixed fermentation was significantly higher than that produced by pure culture with *T. delbrueckii*. However, Li et al. [1] reported that *C. zemplinina* mixed-fermented with *S. cerevisiae* reduced the ethanol and acetic acid concentrations. In our study, only the acetic acid concentration of wines mixed-fermented with F33 and *S. crataegensis* YC30 and *R. lusitaniae* QTX15 increased, whereas it was reduced in other wine samples.

##### Aldehydes

Aldehydes are produced during wine fermentation through alcohol oxidation or acid decarboxylation [62]. A total of six aldehydes were detected in the grape juice and wine samples: acetaldehyde, nonylaldehyde, n-octanal, 1-hexanal, isovaleral, and isobutanal (Table 3, Figure 5A,C). Most aldehydes compounds were found in grape juice, but the contents of aldehyde compounds in wine were much higher than in grape juice, which was mainly attributed to acetaldehyde. The concentration of acetaldehyde in wines was 12–18 times higher than that in grape juice.

Compared with pure fermentation, mixed fermentation with *H. uvarum* QTX22 and *R. lusitaniae* QTX15 significantly increased the content of aldehydes; *S. crataegensis* YC30, *M. pulcherrima* YC12, and *R. lusitaniae* QTX15 significantly increased the contents of nonylaldehyde and isobutanal, which were not detected with pure fermentation. In addition, *P. kluyveri* HSP14 and *R. lusitaniae* QTX15 significantly increased isovaleral and n-octanal concentrations, and *S. crataegensis* YC30 and *H. uvarum* QTX22 significantly increased the concentrations of isovaleral and 1-hexanal.

### 3.5. Relative Odor Activity Values

These substances have the possibility of contributing to the wine’s aroma when the concentrations of aroma compounds in the wine are greater than its odor threshold [5]. The odor activity values (OAVs) of different volatile compounds are shown in Table 4, Figure 6A according to the odor threshold reported in the literature. We report 48 odor activity values, including 1 C13-norisoprenoid (β-damascenone), 8 terpenes, 6 acetate esters, 10 ethyl esters, 3 other esters, 7 alcohols, 5 acids, 6 aldehydes, 1 ketone, and 1 ether compounds. The total OAV of the volatile aroma compounds in wine samples was 2–4 times higher than that in grape juice. Compared with pure fermentation, mixed fermentation with non-*Saccharomyces* yeast strains increased the aroma activity value of acetate esters and ethyl ester substances. Among them, the wine fermented with *S. cerevisiae* F33 and *S. crataegensis* YC30 produced the highest value and the largest amount of active aroma substances, indicated by the increased the OAV values of 3 terpenes (cinnamene, linalool, citronellol), 1 C13-norisoprenoid (β-damascenone), 6 acetate esters (isoamyl ethanoate, benzylcarbinyl acetate, isobutyl acetate, butyl acetate, propyl acetate, and hexyl acetate), 10 ethyl esters (ethyl dodecylate, ethyl heptoate, ethyl caproate, ethyl isobutyrate, etc.), and 3 alcohols (3-octenol, penten-3-ol, 2-heptanol). Notably, the wine produced by mixed fermentation of *S. cerevisiae* F33 and *S. crataegensis* YC30 also showed three lactones (δ-decalactone, γ-nonalactone, and γ-decalactone), which are related to coconut and creamy odors [57], which were not found in other non-*Saccharomyces* strains (Figure 6B). In addition, the mixed fermentation of *S. cerevisiae* F33 and *H. uvarum* QTX22 wine also resulted in a higher total OAV and better aroma characteristics. We found that the aroma activity values of 4 terpenes (naphthalene, cinnamene, citronellol, d-limonene, and α-terpineol), 1 C13-norisoprenoid (β-damascenone), 3 acetate esters (isoamyl ethanoate, benzylcarbinyl acetate, and isobutyl acetate), 6 ethyl ester aroma substances (ethyl dodecylate, ethyl heptoate, ethyl caproate, ethyl butanoate, ethyl phenylethanoate, and ethyl dihydrocinnamate), and 6 alcohols (1-pentanol, 1-heptanol, 1-hexanol, etc.) significantly increased. Simultaneous fermentation with *S. cerevisiae* F33 and *P. kluyveri* HSP14 significantly increased the OAVs of 2-methyl-propanoic acid, butyl acetate, propyl acetate, etc.; and *R. lusitaniae* QTX15 significantly increase the OAVs of n-octanal, nonylaldehyde, p-xylene, benzylcarbinyl acetate, isobutyl acetate, etc. β-damascenone, cinnamene, linalool, citronellol, nerol oxide, benzylcarbinyl acetate, ethyl dodecylate, ethyl dihydrocinnamate, and 1-octanol are mainly associated with floral-like aromas, whereas isoamyl ethanoate, benzylcarbinyl acetate, isobutyl acetate, 1-hexanol, penten-3-ol, 2-heptanol ,and most ethyl esters are related to fruity odors, which contribute to the complexity and typical qualities of wine.

### 3.6. PCA Analysis

To further understand the influence of different fermentation strategies on the aroma compounds of Vidal icewine, principal component analysis was performed on aroma-active substances with an OAV > 1 [72] (Figure 6C). In a 48 × 6 data matrix, the generated data explained 80.78% of the total variance. The Vidal icewine purely fermented with *S. cerevisiae* F33 was located in the third quadrant, which was well-distinguished from the mixed fermented wine, indicating that mixed fermentation with non-*Saccharomyces* yeast strains had a significant effect on improving the aroma substances of Vidal icewine. Wines produced by the mixed fermentation of *S. cerevisiae* F33 with *S. crataegensis* YC30 and *H. uvarum* QTX22 were located in the first and fourth quadrants, respectively, and could also be well-distinguished due to their unique aroma characteristics. Finally, the wine purely fermented with *S. cerevisiae* F33 was located on the farthest negative end of the Y-axis, and was positively correlated with d-limonene, naphthalene, 1-pentanol, and 1-heptanol; the wine produced by mixed fermentation of *S. cerevisiae* F33 with *S. crataegensis* YC30 was located at the farthest positive end of the Y-axis, and was positively correlated with linalool, γ-decalactone, γ-nonalactone, ethyl isobutyrate, δ-decalactone, ethyl propanoate, hexyl acetate, ethyl heptoate, isoamyl ethanoate, ethyl butanoate, and 2-heptanol, being significantly different from the wine produced by pure fermentation in terms of the aroma characteristics. The mixed fermentation with *H. uvarum* QTX22 was located at the farthest point on the positive end of the X-axis and the negative end of the Y-axis, related to d-limonene, 1-pentanol, ethyl heptoate, isoamyl ethanoate, ethyl butanoate, 2-heptanol, etc. Mixed fermentation with *R. lusitaniae* QTX15 was located farthest from the negative end of the X-axis, and was positively correlated with p-xylene, nonylaldehyde, n-octanal, etc.; mixed fermentation with *M. pulcherrima* YC12 and *P. kluyveri* HSP14 showed similar aroma characteristics, both located in the second quadrant on the negative end of the Y-axis. The main characteristic aromas of F33 purely fermented Vidal icewine were alcohols and terpenes, whereas those of mixed fermentation with *S. crataegensis* YC30 and *H. uvarum* QTX22 were esters, further indicating that non-*Saccharomyces* yeast strains can increase the content of ester aroma substances in Vidal icewine. Ester aroma substances are generally related to fruity and floral aroma notes; γ-decalactone and γ-nonalactone have fruity and ice cream odors [57]; hexyl acetate has banana and apple odors [66]; ethyl heptoate has pineapple, banana, and strawberry flavors [52]; and isoamyl ethanoate, ethyl propanoate, and ethyl butanoate [66] all have sweet and fruity odors, which greatly enriches the complexity and quality of the Vidal icewine aroma.

### 3.7. Sensory Analysis

Figure 7A shows the MF value contributed by pure fermentation and mixed fermentation to the aroma of Vidal icewine, including fruity, floral, alcohol, nail polish, caramel, herbal, and overall aromas. In the six wine samples, except for the alcohol note, all others were significant or extremely significant (*p* < 0.05 or *p* < 0.001). Compared with pure fermentation, the overall, fruity, and floral aroma MF values of the wine produced by mixed fermentation with *S. crataegensis* YC30 were the highest, followed by *H. uvarum* QTX22. In addition, *H. uvarum* QTX22, *M. pulcherrima* YC12, and *P. kluyveri* HSP14 also showed strong herbal, caramel and nail polish odors, respectively. To determine whether there was a perceptible difference between the icewine fermented with *S. crataegensis* YC30, *H. uvarum* QTX22, and *S. cerevisiae* F33 (as a control) in the fruity and floral note aspects, triangular tests were also performed (Figure 7B). The wine simultaneously co-fermented with *S. crataegensis* YC30 and *S. cerevisiae* F33 was perceived as having a significantly intense fruity and floral odor compared with that produced by pure fermentation with *S. cerevisiae* F33. No significant differences were found in these odor characteristics with the simultaneous fermentation treatments (*S. cerevisiae* F33 and *H. uvarum* QTX22). Since the aromas of wine are perceived through the perceptual response to odorants, PLSR analysis was used to predict and reveal the correlation between the odor characteristics of wine aroma (fruit, floral, alcohol, nail polish, caramel, and herbal) and the amount of aroma compounds [21] (Figure 7C,D). Only the floral (R^2^ cal/val = 0.993/0.837) and fruity (R^2^ cal/val = 0.995/0.813) aroma notes could be accurately predicted. Among them, the perception of fruity was influenced by the combination of the positive and negative contributions of volatiles. The presence of C13-norisoprenoid (β-damascenone +0.3279), terpenes (+0.2401), ethyl esters (+0.2995), acetate esters (+0.1748), and other esters (+0.1790) positively correlated with this aroma; whereas aldehydes (−0.1384), ketones (−0.1067), and ethers (−0.0941) showed negative correlations. C13-norisoprenoid (β-damascenone, +0.2941), terpenes (+0.1996), ethyl esters (+0.2891), acetate esters (+0.1650), and other esters (+0.1921) positively correlated with floral aroma notes, and aldehydes (−0.0976), ketones (−0.0660), and ethers (−0.0442) showed negative correlation, which is consistent with a previous study in which aroma characteristics were affected by the complex contributions of various aroma-active substances [32]. These results further illustrate that the aroma characteristics and quality of wine are affected by the type and diversity of volatile aroma substances, especially esters and terpenes [10].

The relationships between the 48 major aroma substances (OAV > 1) and sensory analysis are depicted in Figure 4E. Wine samples’ fruity and floral odors were positively correlated with β-damascenone, cinnamene, isoamyl ethanoate, ethyl dodecylate, ethyl caproate, ethyl butanoate, ethyl dihydrocinnamate, and n-decanoic acid (correlation coefficient > 0.8), which further verified that terpenes and esters were related to the production of floral and fruity odors in Vidal icewine.

## 4. Conclusions

We analyzed the effects of commercial *S. cerevisiae* F33 in pure culture as well as mixed culture with five indigenous non-*Saccharomyces* yeast strains to evaluate the effects on the aroma characteristics of Vidal icewine. In the process of pure fermentation with F33, the maximum biomass was reached on day 3, whereas co-inoculation with non-*Saccharomyces* yeast strains reached maximum biomass on day 4; the non-*Saccharomyces* yeast strains were undetectable on days 5 and 7. Simultaneous fermentation increased the concentrations of total polyphenol, especially the *S. crataegensis* YC30 strain, which significantly increased the concentrations of myricetin, chlorogenic acid, quercetin, vanillic acid, p-hydroxybenzonic acid, etc. A total of 118 aroma compounds were detected, mainly including acetate esters, ethyl esters, alcohols, and acids. Simultaneous fermentation markedly increased the total content of volatile aroma compounds and the contents of acetate, ethyl ester, other esters, and terpenes in Vidal icewine, especially with *S. crataegensis* YC30 and *H. uvarum* QTX22. *S. crataegensis* YC30 significantly increased the aroma activity values of 1 C13-norisoprenoid (β-damascenone), 6 acetate esters, 10 ethyl esters, 3 terpenes, and 3 alcohols, and produced obvious floral and fruity aroma notes. PLSR analysis showed that this was mainly related to C13-norisoprenoids (β-damascenone, r = 0.3279), terpenes (r = 0.2401), ethyl esters (r = 0.2995), acetate esters (r = 0.1748), and other esters (r = 0.1790). This study provides a new method for the development and production of icewine.

## Figures and Tables

**Figure 1 foods-10-01452-f001:**
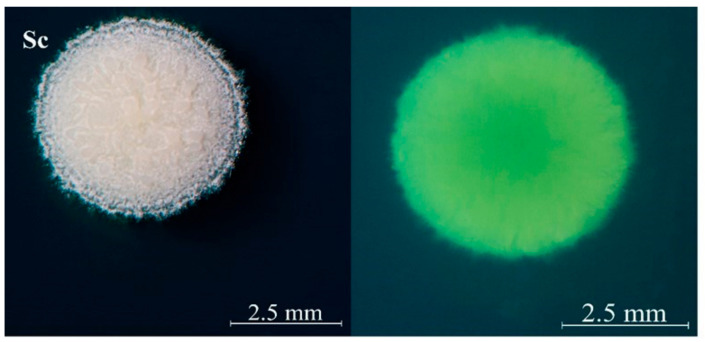
The colony morphology diagram of *S. crataegensis* YC30.

**Figure 2 foods-10-01452-f002:**
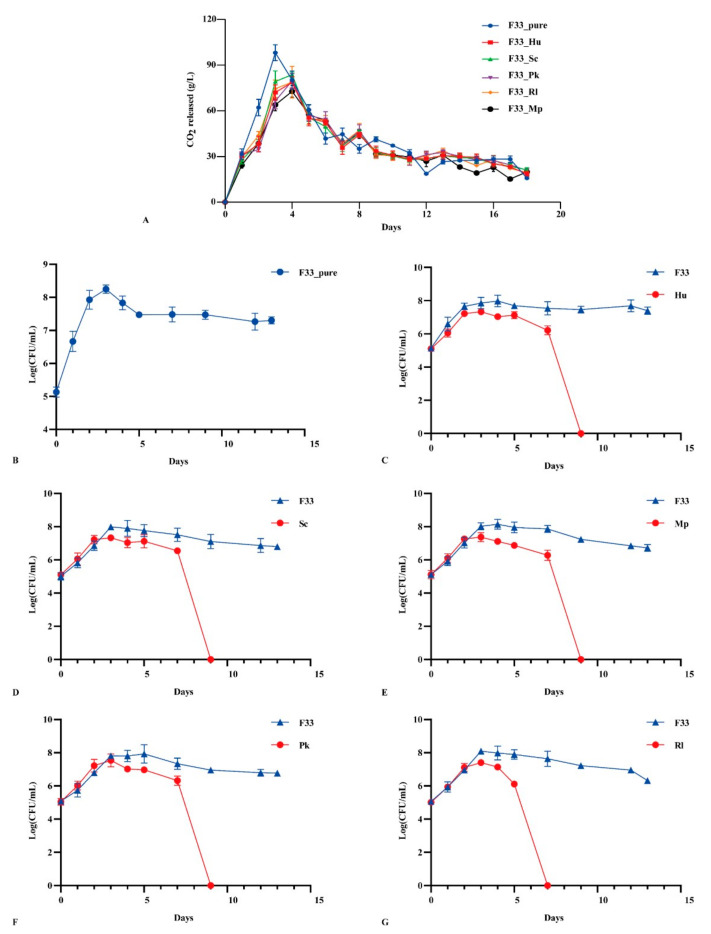
The fermentation kinetics characteristics and yeast population dynamics in Vidal icewine during fermentation with six fermentation strategies. (**A**) CO_2_ release in pure and co-mixed fermentation. pure_F33: *S. cerevisiae* F33 monoculture. F33_Hu: co-inoculation of *S. cerevisiae* F33 and *H. uvarum* QTX22. F33_Sc: co-inoculation of *S. cerevisiae* F33 and *S. crataegensis* Y30. F33_Mp: co-inoculation of *S. cerevisiae* F33 and *M. pulcherrima* YC15. F33_Pk: co-inoculation of *S. cerevisiae* F33 and *P. kluyveri* HSP11. F33_Rl: co-inoculation of *S. cerevisiae* F33 and *R. lusitaniae* QTX26. (**B**–**G**) Yeast population dynamics in different fermentations. Hu: *H. uvarum* QTX22. Sc: *S. crataegensis*. Mp: *M. pulcherrima* YC15. Pk: *P. kluyveri* HSP11. Rl: *R. lusitaniae* QTX26.

**Figure 3 foods-10-01452-f003:**
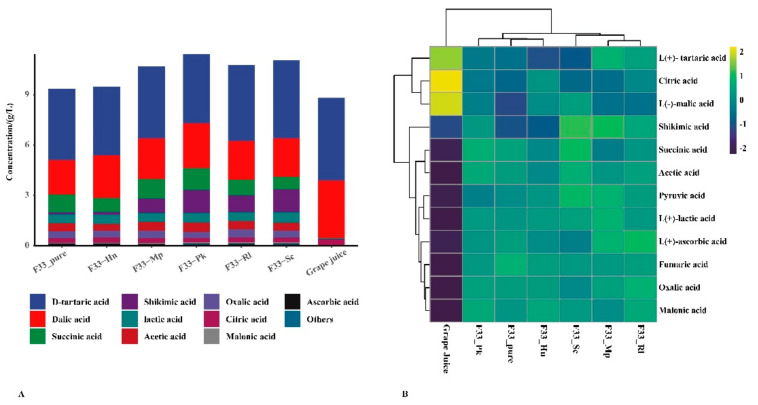
The distribution of organic acids content under different fermentation strategies. (**A**) Stacked graph. (**B**) Clustering pheatmap.

**Figure 4 foods-10-01452-f004:**
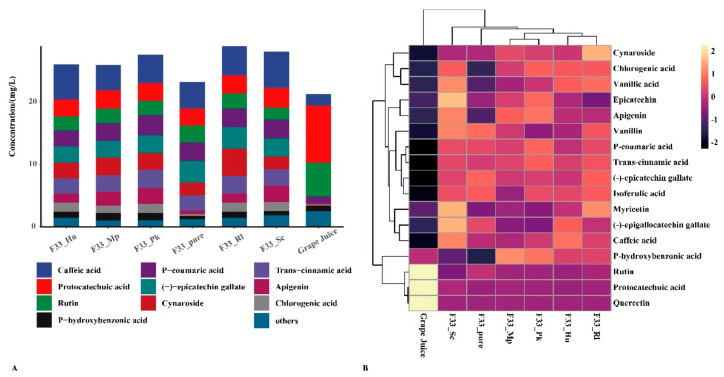
The distribution of polyphenols content under different fermentation strategies. (**A**) Stacked graph. (**B**) Clustering pheatmap.

**Figure 5 foods-10-01452-f005:**
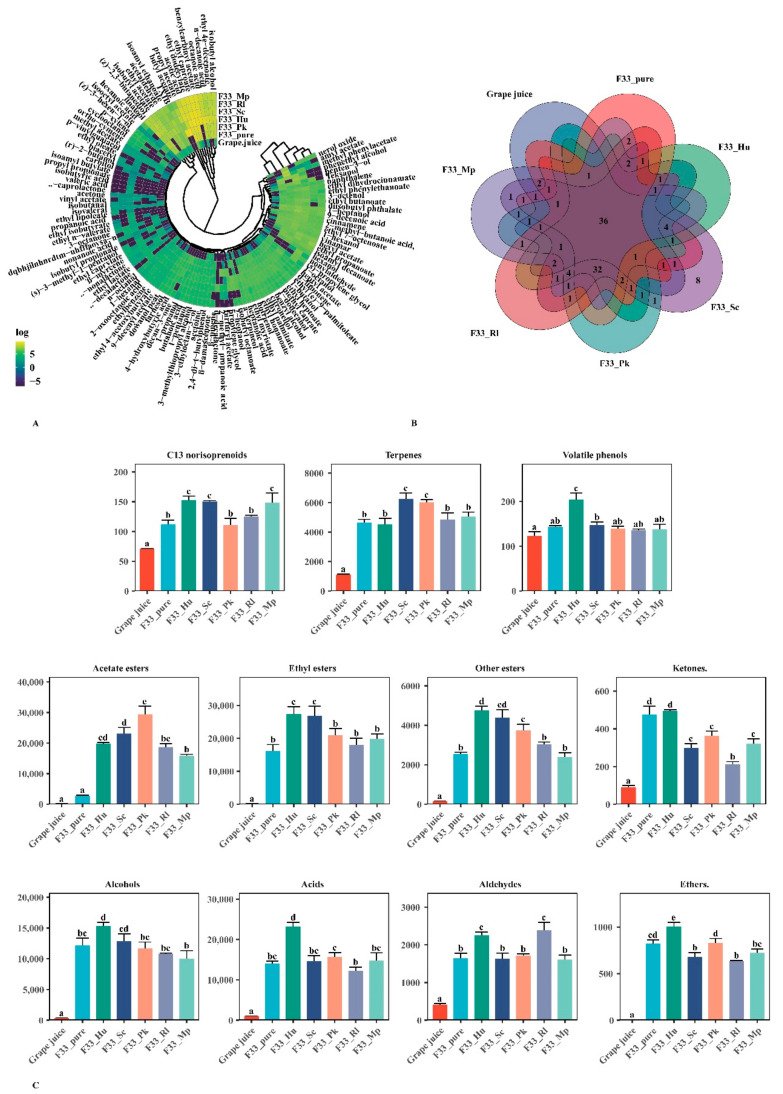
Volatile compounds from Vidal grape juice and icewines fermented with six fermentation strategies. pure_F33: *S. cerevisiae* F33 monoculture. F33_Hu: co-inoculation of *S. cerevisiae* F33 and *H. uvarum* QTX22. F33_Sc: co-inoculation of *S. cerevisiae* F33 and *S. crataegensis* Y30. F33_Mp: co-inoculation of *S. cerevisiae* F33 and *M. pulcherrima* YC15. F33_Pk: co-inoculation of *S. cerevisiae* F33 and *P. kluyveri* HSP11. F33_Rl: co-inoculation of *S. cerevisiae* F33 and *R. lusitaniae* QTX26. (**A**) Pheatmap clustering graph of all the individual volatile substances. Yellow: relatively high production; Black: relatively low production. Count represents the standardized content aroma-active compounds. (**B**) Venn diagram showing the distribution of the numbers of the aroma compounds. (**C**) Column diagram showing the content distribution of the aroma categories. Different letters indicate statistically significant differences.

**Figure 6 foods-10-01452-f006:**
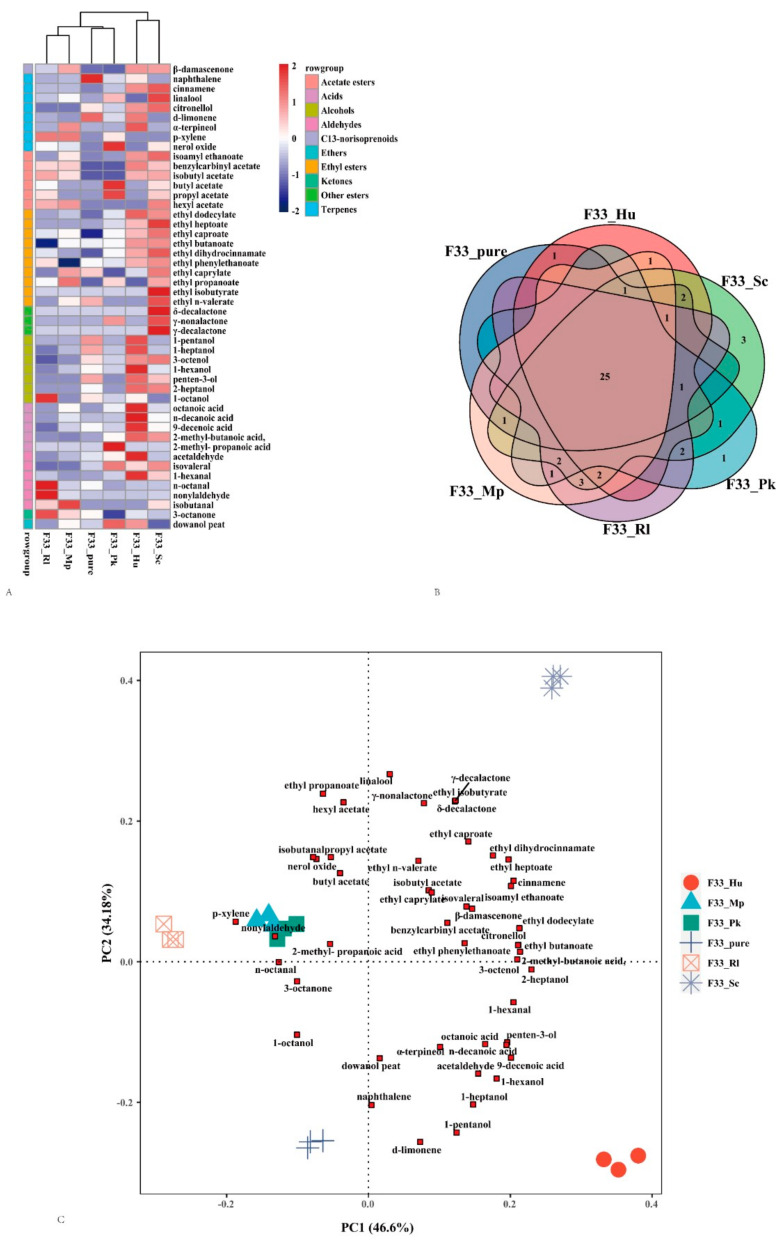
Aroma-active compounds from Vidal icewine fermented with six fermentation strategies. pure_F33: *S. cerevisiae* F33 monoculture. F33_Hu: co-inoculation of *S. cerevisiae* F33 and *H. uvarum* QTX22. F33_Sc: co-inoculation of *S. cerevisiae* F33 and *S. crataegensis* Y30. F33_Mp: co-inoculation of *S. cerevisiae* F33 and *M. pulcherrima* YC15. F33_Pk: co-inoculation of *S. cerevisiae* F33 and *P. kluyveri* HSP11. F33_Rl: co-inoculation of *S. cerevisiae* F33 and *R. lusitaniae* QTX26. (**A**) Pheatmap clustering graph of aroma-active compounds. Red: relatively high production; blue: relatively low production. Count represents the standardized content aroma-active compounds. (**B**) Venn diagrams of the aroma-active compounds. (**C**) Principal component analysis of aroma-active compounds.

**Figure 7 foods-10-01452-f007:**
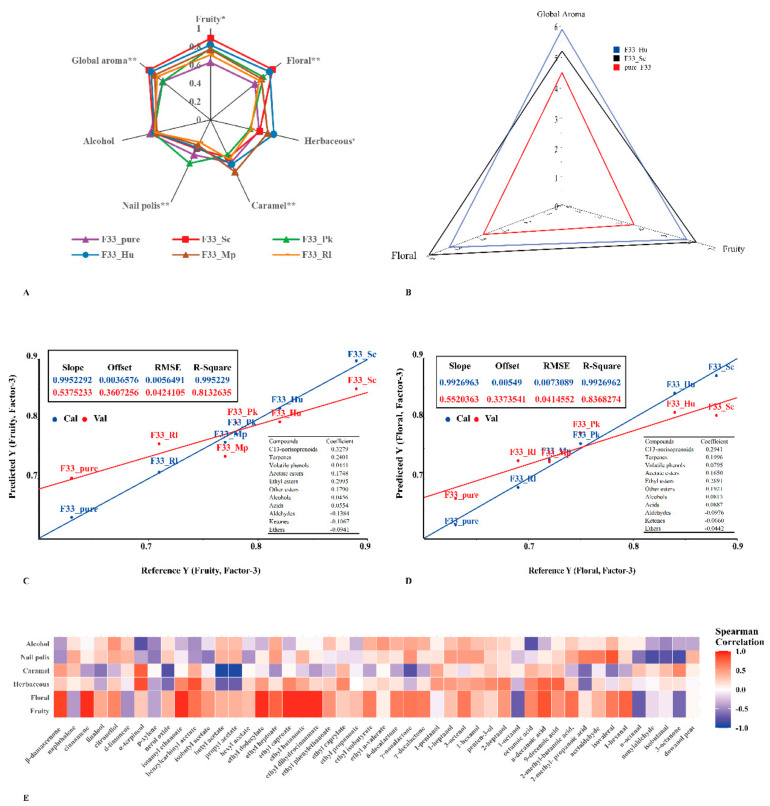
The sensory evaluation of Vidal icewine fermented with six fermentation strategies. pure_F33: *S. cerevisiae* F33 monoculture. F33_Hu: co-inoculation of *S. cerevisiae* F33 and *H. uvarum* QTX22. F33_Sc: co-inoculation of *S. cerevisiae* F33 and *S. crataegensis* Y30. F33_Mp: co-inoculation of *S. cerevisiae* F33 and *M. pulcherrima* YC15. F33_Pk: co-inoculation of *S. cerevisiae* F33 and *P. kluyveri* HSP11. F33_Rl: co-inoculation of *S. cerevisiae* F33 and *R. lusitaniae* QTX26. (**A**) Modified frequency (MF) values of aroma characteristics in six fermentation strategies. * and **, significant at *p* < 0.05 and 0.01, respectively. (**B**) The analysis results of triangular tests. Grades ranked from 0 (poorly intense) to 7 (very intense). (**C**,**D**) PLS regression of fruity (**C**) and floral (**D**) aroma volatiles. Val, validation; Cal, calibration. (**E**) Correlation analysis between major aroma substances (OAV > 1) and sensory analysis.

**Table 1 foods-10-01452-t001:** Physicochemical parameters of Vidal grape juice and icewines fermented with six fermentation strategies.

	Grape Juice	F33_Pure	F33_Hu	F33_Sc	F33_Mp	F33_Pk	F33_Rl
Ethanol (%)	ND	10.12 ± 0.93 ^c^	8.66 ± 0.53 ^bc^	8.46 ± 0.17 ^b^	8.21 ± 0.08 ^b^	7.35 ± 0.70 ^b^	7.82 ± 0.56 ^b^
TS (g/L) **	71.93 ± 6.86 ^c^	344.0 ± 6.19 ^d^	36.76 ± 3.01 ^ab^	36.76 ± 3.62 ^ab^	46.65 ± 4.07 ^b^	32.36 ± 2.53 ^a^	45.55 ± 6.03 ^ab^
TA (g/L)	6.14 ± 0.37 ^a^	6.09 ± 0.52 ^a^	5.97 ± 0.21 ^a^	6.49 ± 0.11 ^a^	6.66 ± 0.23 ^a^	6.31 ± 0.39 ^a^	6.14 ± 0.11 ^a^
PH	3.77 ± 0.44 ^a^	3.64 ± 0.28 ^a^	3.87 ± 0.39 ^a^	3.84 ± 0.23 ^a^	3.90 ± 0.08 ^a^	3.89 ± 0.35 ^a^	3.89 ± 0.22 ^a^
Organic acids (g/L)							
Citric acid **	0.358 ± 0.033 ^b^	0.274 ± 0.003 ^a^	0.299 ± 0.024 ^a^	0.279 ± 0.022 ^a^	0.284 ± 0.005 ^a^	0.276 ± 0.006 ^a^	0.293 ± 0.023 ^a^
Pyruvic acid **	ND	0.047 ± 0.003 ^bc^	0.051 ± 0.005 ^c^	0.065 ± 0.006 ^d^	0.041 ± 0.002 ^b^	0.069 ± 0.002 ^d^	0.055 ± 0.002 ^c^
Oxalic acid **	ND	0.41 ± 0.03 ^bc^	0.403 ± 0.05 ^bc^	0.423 ± 0.018 ^bc^	0.418 ± 0.029 ^bc^	0.348 ± 0.032 ^b^	0.47 ± 0.033 ^c^
Fumaric acid **	0.007 ± 0.001 ^a^	0.024 ± 0.001 ^b^	0.022 ± 0.002 ^b^	0.021 ± 0.002 ^b^	0.021 ± 0.001 ^b^	0.021 ± 0.001 ^b^	0.022 ± 0.001 ^b^
Ascorbic acid **	0.022 ± 0.001 ^a^	0.057 ± 0.005 ^bd^	0.052 ± 0.002 ^b^	0.063 ± 0.002 ^cd^	0.054 ± 0.004 ^bc^	0.048 ± 0.004 ^b^	0.066 ± 0.006 ^d^
Tartaric acid **	4.938 ± 0.198 ^c^	4.258 ± 0.043 ^ab^	4.102 ± 0.142 ^a^	4.654 ± 0.123 ^bc^	4.319 ± 0 ^ab^	4.125 ± 0.23 ^a^	4.554 ± 0.164 ^bc^
Malic acid **	3.412 ± 0.329 ^c^	2.035 ± 0.113 ^a^	2.548 ± 0.234 ^ab^	2.296 ± 0.204 ^ab^	2.419 ± 0.211 ^ab^	2.699 ± 0.282 ^b^	2.296 ± 0.143 ^ab^
Malonic acid **	ND	0.062 ± 0.003 ^bc^	0.07 ± 0.004 ^cd^	0.058 ± 0.004 ^b^	0.072 ± 0.006 ^cd^	0.064 ± 0.001 ^bd^	0.073 ± 0.005 ^d^
Succinic acid **	0.045 ± 0.003 ^a^	1.064 ± 0.092 ^de^	0.842 ± 0.03 ^bc^	0.76 ± 0.03 ^b^	1.156 ± 0.064 ^ef^	1.287 ± 0.089 ^f^	0.943 ± 0.081 ^cd^
Shikimic acid **	0.048 ± 0.003 ^a^	0.124 ± 0.003 ^ab^	0.16 ± 0.009 ^b^	1.347 ± 0.036 ^e^	0.854 ± 0.043 ^c^	1.378 ± 0.024 ^e^	0.992 ± 0.075 ^d^
Lactic acid **	ND	0.519 ± 0 ^b^	0.538 ± 0.034 ^b^	0.641 ± 0.061 ^c^	0.542 ± 0.03 ^b^	0.569 ± 0.025 ^bc^	0.527 ± 0.011 ^b^
Acetic acid **	ND	0.492 ± 0.026 ^bd^	0.425 ± 0.038 ^b^	0.468 ± 0.012 ^bc^	0.543 ± 0.042 ^cd^	0.564 ± 0.041 ^d^	0.512 ± 0.022 ^cd^

Data are mean values of three independent experiments ± standard deviation. Mean values displaying different letters within each row are significantly different according to the Duncan test at 95% confidence level. pure_F33: fermentation with *S. cerevisiae* F33 in pure. F33_ Hu, F33_ Sc, F33_ Mp, F33_ Pk, F33_ Rl represented the simultaneous mixed fermentation of F33 with *H. uvarum* QTX22, *S. crataegensis* YC30, *M. pulcherrima* YC15, *P. kluyveri* HSP11 and *R. lusitaniae* QTX26, respectively. ND: not detected. TA = total acid (expressed as percentage of tartaric acid). TS = total sugar. **, significant at *p* < 0.01.

**Table 2 foods-10-01452-t002:** Concentrations of polyphenols (mg/L) of Vidal grape juice and icewines fermented with six fermentation strategies.

	Grape Juice	F33_pure	F33_Hu	F33_Sc	F33_Mp	F33_Pk	F33_Rl
Protocatechuic acid **	9.124 ± 0.790 ^b^	2.715 ± 0.144 ^a^	2.667 ± 0.148 ^a^	3.124 ± 0.143 ^a^	2.908 ± 0.324 ^a^	2.869 ± 0.224 ^a^	2.897 ± 0.161 ^a^
P-hydroxybenzonic acid *	0.867 ± 0.065 ^b^	0.522 ± 0.019 ^a^	0.956 ± 0.066 ^b^	0.663 ± 0.023 ^a^	1.186 ± 0.059 ^c^	1.122 ± 0.074 ^c^	0.966 ± 0.017 ^b^
Chlorogenic acid **	0.200 ± 0.017 ^a^	0.308 ± 0.033 ^a^	1.434 ± 0.112 ^bc^	1.456 ± 0.064 ^c^	1.175 ± 0.082 ^b^	1.506 ± 0.153 ^c^	1.434 ± 0.129 ^bc^
Vanillic acid **	0.176 ± 0.006 ^a^	0.233 ± 0.012 ^a^	0.548 ± 0.019 ^c^	0.630 ± 0.069 ^c^	0.365 ± 0.048 ^b^	0.426 ± 0.041 ^b^	0.576 ± 0.027 ^c^
(-)-epigallocatechin gallate **	ND	0.295 ± 0.023 ^d^	0.352 ± 0.010 ^e^	0.488 ± 0.021 ^f^	0.125 ± 0.011 ^b^	0.114 ± 0.011 ^b^	0.221 ± 0.013 ^c^
Epicatechin **	0.017 ± 0.001 ^a^	0.029 ± 0.001 ^bc^	0.033 ± 0.001 ^cd^	0.066 ± 0.005 ^f^	0.041 ± 0.004 ^d^	0.052 ± 0.004 ^e^	0.024 ± 0.002 ^ab^
Vanillin **	0.028 ± 0.001 ^a^	0.285 ± 0.024 ^de^	0.170 ± 0.015 ^b^	0.304 ± 0.014 ^e^	0.211 ± 0.015 ^c^	0.148 ± 0.012 ^b^	0.254 ± 0.009 ^d^
P-coumaric acid **	0.966 ± 0.029 ^a^	2.941 ± 0.284 ^bc^	2.580 ± 0.129 ^b^	2.985 ± 0.136 ^bc^	2.810 ± 0.296 ^bc^	3.261 ± 0.000 ^c^	2.947 ± 0.179 ^bc^
(-)-epicatechin gallate **	ND	3.497 ± 0.334 ^c^	2.610 ± 0.188 ^b^	2.908 ± 0.126 ^bc^	2.708 ± 0.097 ^b^	2.838 ± 0.327 ^b^	3.436 ± 0.206 ^c^
Isoferulic acid **	0.038 ± 0.002 ^a^	0.294 ± 0.039 ^c^	0.262 ± 0.011 ^c^	0.276 ± 0.000 ^c^	0.185 ± 0.021 ^b^	0.272 ± 0.026 ^c^	0.292 ± 0.005 ^c^
Rutin **	5.400 ± 0.389 ^c^	2.700 ± 0.258 ^b^	2.256 ± 0.135 ^ab^	1.924 ± 0.126 ^a^	2.317 ± 0.184 ^ab^	2.279 ± 0.046 ^ab^	2.428 ± 0.170 ^ab^
Trans-cinnamic acid **	0.012 ± 0.001 ^a^	2.427 ± 0.146 ^b^	2.487 ± 0.149 ^bc^	2.637 ± 0.147 ^bc^	2.577 ± 0.220 ^bc^	2.966 ± 0.136 ^c^	2.787 ± 0.348 ^bc^
Apigenin **	0.300 ± 0.019 ^a^	0.582 ± 0.047 ^b^	1.477 ± 0.074 ^c^	2.615 ± 0.069 ^f^	2.278 ± 0.060 ^d^	2.466 ± 0.025 ^e^	1.469 ± 0.025 ^c^
Myricetin **	0.032 ± 0.001 ^a^	0.036 ± 0.001 ^ab^	0.046 ± 0.001 ^b^	0.068 ± 0.004 ^c^	0.041 ± 0.005 ^ab^	0.037 ± 0.003 ^ab^	0.063 ± 0.007 ^c^
Cynaroside *	ND	2.048 ± 0.182 ^b^	2.461 ± 0.186 ^bc^	2.044 ± 0.108 ^b^	2.895 ± 0.181 ^c^	2.659 ± 0.027 ^bc^	4.422 ± 0.585 ^d^
Caffeic acid *	1.904 ± 0.148 ^a^	4.267 ± 0.128 ^b^	5.632 ± 0.627 ^c^	5.873 ± 0.411 ^c^	4.114 ± 0.179 ^b^	4.568 ± 0.121 ^b^	4.741 ± 0.246 ^b^
Quercetin **	2.228 ± 0.194 ^b^	0.021 ± 0.001 ^a^	0.021 ± 0.001 ^a^	0.023 ± 0.001 ^a^	0.019 ± 0.002 ^a^	0.022 ± 0.002 ^a^	0.022 ± 0.002 ^a^

Data are mean values of three independent experiments ± standard deviation. Mean values displaying different letters within each row are significantly different according to the Duncan test at 95% confidence level. pure_F33: fermentation with *S. cerevisiae* F33 in pure. F33_ Hu, F33_ Sc, F33_ Mp, F33_ Pk, F33_ Rl represented the simultaneous mixed fermentation of F33 with *H. uvarum* QTX22, *S. crataegensis* YC30, *M. pulcherrima* YC15, *P. kluyveri* HSP11 and *R. lusitaniae* QTX26, respectively. ND: not detected. * and **, significant at *p* < 0.05 and 0.01, respectively.

**Table 3 foods-10-01452-t003:** All the individual volatile compounds (μg/L) from Vidal icewine fermented with six fermentation strategies.

Compounds	CAS	RI	Formula	Grape Juice	F33_Hu	F33_Sc	F33_Mp	F33_Pk	F33_pure	F33_Rl
**C13-norisoprenoids**										
β-damascenone	23726-93-4	1440	C_13_H_18_O	70.742 ± 6.288 ^a^	152.303 ± 4.030 ^c^	150.536 ± 12.322 ^c^	148.687 ± 7.868 ^c^	110.502 ± 9.045 ^b^	112.345 ± 2.972 ^b^	125.335 ± 9.948 ^b^
**Terpenes**										
ethylbenzene	100-41-4	893	C_8_H_10_	2.435 ± 0.199 ^a^	112.692 ± 9.224 ^d^	85.223 ± 2.557 ^bc^	89.754 ± 7.180 ^bc^	74.833 ± 9.196 ^b^	101.783 ± 9.160 ^cd^	74.109 ± 6.792 ^b^
naphthalene	91-20-3	1231	C_10_H_8_	148.192 ± 5.343 ^a^	416.705 ± 30.049 ^c^	288.905 ± 23.648 ^b^	305.452 ± 21.382 ^b^	342.680 ± 13.707 ^bc^	686.057 ± 74.208 ^d^	295.368 ± 36.891 ^b^
cinnamene	100-42-5	883	C_8_H_8_	383.869 ± 36.619 ^a^	752.324 ± 71.767 ^c^	837.172 ± 52.281 ^c^	501.109 ± 13.258 ^b^	516.951 ± 18.639 ^b^	433.728 ± 11.475 ^ab^	510.919 ± 18.421 ^b^
linalool	78-70-6	1082	C_10_H_18_O	182.614 ± 15.920 ^a^	1922.437 ± 183.389 ^b^	3306.532 ± 288.257 ^e^	2469.939 ± 154.248 ^cd^	2771.142 ± 264.350 ^d^	2102.666 ± 55.631 ^bc^	2255.023 ± 112.751 ^bc^
citronellol	106-22-9	1179	C_10_H_20_O	ND	392.406 ± 20.390 ^de^	423.152 ± 29.621 ^e^	248.001 ± 19.684 ^b^	296.777 ± 20.774 ^bc^	339.634 ± 27.800 ^cd^	250.362 ± 27.879 ^b^
farnesol	4602-84-0	1710	C_15_H_26_O	ND	177.063 ± 11.611 ^cd^	143.529 ± 2.486 ^b^	138.131 ± 16.285 ^b^	197.755 ± 8.620 ^d^	198.741 ± 5.962 ^d^	163.259 ± 13.363 ^bc^
d-limonene	5989-27-5	1018	C_10_H_16_	99.847 ± 4.992 ^a^	336.024 ± 36.347 ^c^	190.037 ± 10.581 ^b^	198.335 ± 17.176 ^b^	232.202 ± 16.744 ^b^	346.862 ± 9.177 ^c^	209.016 ± 9.578 ^b^
β-myrcene	123-35-3	958	C_10_H_16_	14.564 ± 1.137 ^b^	ND	52.174 ± 3.421 ^c^	52.005 ± 4.766 ^c^	58.626 ± 5.111 ^cd^	63.110 ± 5.466 ^d^	52.099 ± 2.271 ^c^
β-pinene	127-91-3	943	C_10_H_16_	ND	80.024 ± 6.550 ^b^	135.067 ± 8.104 ^c^	126.075 ± 14.540 ^c^	134.699 ± 8.193 ^c^	147.793 ± 12.097 ^c^	ND
α-terpineol	98-55-5	1143	C_10_H_18_O	58.715 ± 3.667 ^b^	346.417 ± 18.331 ^d^	248.798 ± 32.913 ^c^	264.095 ± 13.205 ^c^	ND	ND	245.392 ± 13.663 ^c^
p-xylene	106-42-3	907	C_8_H_10_	ND	ND	ND	200.713 ± 14.050 ^c^	104.695 ± 3.775 ^b^	ND	205.178 ± 11.424 ^c^
ortho-cymene	527-84-4	134	C_10_H_14_	ND	ND	ND	104.903 ± 6.551 ^b^	ND	180.368 ± 10.822 ^c^	100.813 ± 2.667 ^b^
2,6-Di(tert-butyl)-4-hydroxy-4 -methyl-2,5-cyclohexadien-1-one	10396-80-2	236	C_15_H_24_O2	12.057 ± 0.435 ^b^	ND	44.632 ± 2.715 ^d^	ND	ND	36.560 ± 1.900 ^c^	62.032 ± 1.861 ^e^
nerol oxide	1786-08-9	1125	C_10_H_16_O	200.123 ± 9.171 ^b^	ND	506.272 ± 43.256 ^e^	346.800 ± 15.892 ^c^	1289.712 ± 38.691 ^f^	ND	420.144 ± 19.253 ^d^
**Volatile phenols**										
2,4-di-t-butylphenol	96-76-4	1555	C_14_H_22_O	122.009 ± 4.399 ^a^	168.209 ± 8.901 ^b^	118.566 ± 1.186 ^a^	113.006 ± 4.926 ^a^	107.611 ± 6.546 ^a^	110.456 ± 7.653 ^a^	110.801 ± 3.995 ^a^
p-vinylguaiacol	7786-61-0	1293	C_9_H_10_O_2_	0.760 ± 0.042 ^a^	35.550 ± 2.220 ^d^	28.858 ± 2.020 ^bc^	24.896 ± 1.743 ^b^	31.405 ± 1.570 ^cd^	32.540 ± 1.127 ^cd^	24.964 ± 2.176 ^b^
**Acetate esters**										
ethyl acetate	141-78-6	586	C_4_H_8_O_2_	36.591 ± 3.354 ^a^	1793.153 ± 111.982 ^d^	1147.773 ± 79.520 ^bc^	1239.243 ± 54.017 ^bc^	1218.622 ± 76.103 ^bc^	1319.687 ± 117.296 ^c^	1043.714 ± 83.497 ^b^
isoamyl ethanoate	123-92-2	820	C_7_H_14_O2	36.070 ± 1.443 ^a^	1653.245 ± 103.245 ^d^	1856.324 ± 115.927 ^d^	1255.537 ± 45.269 ^c^	789.373 ± 91.035 ^b^	704.055 ± 30.689 ^b^	779.904 ± 71.479 ^b^
methyl acetate	79-20-9	487	C_3_H_6_O_2_	ND	29.858 ± 1.368 ^c^	18.378 ± 1.753 ^b^	17.119 ± 1.522 ^b^	17.138 ± 0.453 ^b^	18.705 ± 0.857 ^b^	16.041 ± 0.976 ^b^
9-decenyl acetate	50816-18-7	1371	C_12_H_22_O_2_	ND	85.289 ± 7.386 ^c^	62.297 ± 3.469 ^b^	62.817 ± 4.743 ^b^	88.262 ± 2.335 ^c^	60.878 ± 4.983 ^b^	55.911 ± 4.438 ^b^
heptyl acetate	112-06-1	1086	C_9_H_18_O_2_	23.569 ± 1.650 ^a^	565.351 ± 46.276 ^d^	220.310 ± 11.015 ^c^	198.468 ± 1.985 ^bc^	192.093 ± 16.412 ^bc^	209.375 ± 6.281 ^c^	154.255 ± 8.589 ^b^
benzylcarbinyl acetate	103-45-7	1259	C_10_H_12_O_2_	ND	8355.253 ± 521.785 ^c^	6552.383 ± 472.499 ^b^	6272.270 ± 287.432 ^b^	ND	ND	5981.076 ± 467.137 ^b^
isobutyl acetate	110-19-0	721	C_6_H_12_O_2_	55.399 ± 2.216 ^a^	2066.293 ± 94.689 ^c^	2166.231 ± 156.209 ^c^	1666.309 ± 88.173 ^b^	ND	ND	2123.640 ± 212.364 ^c^
butyl acetate	123-86-4	774	C_6_H_12_O_2_	ND	1027.800 ± 17.802 ^b^	3346.875 ± 57.970 ^d^	712.800 ± 72.341 ^b^	8517.500 ± 596.225 ^e^	ND	2125.800 ± 221.940 ^c^
propyl acetate	109-60-4	666	C_5_H_10_O_2_	ND	3471.600 ± 125.170 ^b^	6046.400 ± 742.994 ^c^	3319.200 ± 87.818 ^b^	12370.400 ± 1237.040 ^d^	ND	5502.000 ± 524.857 ^c^
hexyl acetate	142-92-7	984	C_8_H_16_O_2_	ND	668.210 ± 78.781 ^c^	993.560 ± 69.549 ^e^	892.314 ± 70.825 ^de^	663.210 ± 56.665 ^c^	332.569 ± 25.974 ^b^	801.792 ± 28.909 ^cd^
amyl acetate	628-63-7	884	ND	ND	6.789 ± 0.413 ^a^	11.352 ± 0.341 ^a^	ND	2411.600 ± 63.805 ^b^	ND	24.650 ± 2.219 ^a^
isooctyl acetate	31565-19-2	1523	C_10_H_20_O_2_	ND	9.365 ± 0.487 ^a^	12.457 ± 0.374 ^a^	12.540 ± 0.783 ^a^	325.652 ± 34.310 ^b^	2.366 ± 0.103 ^a^	10.325 ± 0.723 ^a^
methyl phenylacetate	101-41-7	1160	ND	ND	124.560 ± 5.708 ^a^	532.140 ± 24.386 ^b^	121.250 ± 7.951 ^a^	2258.650 ± 148.110 ^c^	33.456 ± 1.004 ^a^	55.410 ± 4.831 ^a^
furfuryl acetate	623-17-6	1009	ND	ND	12.365 ± 0.567 ^a^	114.021 ± 12.697 ^b^	ND	563.256 ± 29.805 ^c^	9.632 ± 0.674 ^a^	25.364 ± 1.342 ^a^
**Ethyl esters**										
ethyl dodecylate	106-33-2	1580	C_14_H_28_O_2_	123.400 ± 3.265 ^a^	9751.865 ± 258.010 ^e^	8865.334 ± 386.431 ^e^	5740.305 ± 604.779 ^cd^	6228.243 ± 186.847 ^d^	4468.127 ± 389.522 ^b^	5271.596 ± 139.473 ^bc^
ethyl heptoate	106-30-9	1099	C_9_H_18_O_2_	ND	342.760 ± 17.810 ^c^	456.321 ± 31.615 ^d^	213.880 ± 18.274 ^b^	256.669 ± 27.042 ^b^	217.832 ± 27.207 ^b^	217.230 ± 16.966 ^b^
ethyl caproate	123-66-0	987	C_8_H_16_O_2_	14.016 ± 0.853 ^a^	7896.235 ± 753.253 ^cd^	8865.336 ± 845.699 ^d^	6961.214 ± 638.006 ^c^	6831.520 ± 361.490 ^c^	4447.635 ± 117.673 ^b^	6608.752 ± 302.851 ^c^
ethyl butanoate	105-54-4	785	C_6_H_12_O_2_	3.078 ± 0.268 ^a^	800.858 ± 52.516 ^d^	819.353 ± 40.968 ^d^	506.172 ± 22.064 ^c^	510.596 ± 22.256 ^c^	476.131 ± 50.164 ^c^	217.327 ± 23.508 ^b^
ethyl myristate	124-06-1	1779	C_16_H_32_O_2_	ND	331.083 ± 14.432 ^d^	426.836 ± 50.323 ^e^	262.372 ± 15.959 ^c^	398.082 ± 31.091 ^de^	167.334 ± 6.693 ^b^	146.116 ± 6.369 ^b^
ethyl nonanoate	123-29-5	1282	C_11_H_22_O_2_	ND	386.235 ± 17.699 ^c^	456.321 ± 31.942 ^d^	316.691 ± 27.426 ^b^	319.467 ± 8.452 ^b^	277.837 ± 24.221 ^b^	322.431 ± 30.758 ^bc^
ethyl palmitate	628-97-7	1978	C_18_H_36_O_2_	ND	398.775 ± 3.988 ^de^	445.478 ± 16.062 ^e^	283.855 ± 22.170 ^b^	342.940 ± 30.481 ^c^	355.733 ± 17.787 ^cd^	273.505 ± 9.861 ^b^
ethyl lactate	97-64-3	848	C_5_H_10_O_3_	ND	59.112 ± 4.617 ^c^	69.336 ± 2.773 ^d^	36.924 ± 1.692 ^b^	38.662 ± 2.679 ^b^	38.125 ± 4.124 ^b^	40.215 ± 3.836 ^b^
ethyl dihydrocinnamate	2021-28-5	1359	C_11_H_14_O_2_	ND	642.765 ± 17.006 ^e^	782.340 ± 28.208 ^f^	347.334 ± 26.223 ^c^	477.504 ± 21.882 ^d^	225.021 ± 25.951 ^b^	443.101 ± 13.293 ^d^
ethyl 4e-decenoate	76649-16-6	1389	C_12_H_22_O_2_	ND	4446.576 ± 320.647 ^e^	3007.235 ± 104.174 ^bc^	2807.420 ± 202.446 ^bc^	3355.207 ± 146.250 ^cd^	3792.373 ± 330.611 ^d^	2687.685 ± 123.165 ^b^
ethyl (e)-9-palmitoleate	54546-22-4	1966	C_18_H_34_O_2_	12.300 ± 0.325 ^a^	392.378 ± 23.867 ^d^	266.860 ± 25.457 ^c^	210.950 ± 12.832 ^b^	277.847 ± 15.470 ^c^	267.319 ± 21.881 ^c^	190.610 ± 7.624 ^b^
ethyl 4-acetoxybutyrate	25560-91-2	1151	ND	ND	92.067 ± 4.013 ^d^	74.297 ± 7.087 ^bc^	63.795 ± 5.847 ^b^	65.518 ± 2.362 ^b^	85.816 ± 7.332 ^cd^	65.822 ± 3.665 ^b^
ethyl 7-octenoate	35194-38-8	1173	C_10_H_18_O_2_	ND	794.572 ± 78.256 ^e^	649.356 ± 19.481 ^cd^	519.541 ± 22.646 ^b^	742.910 ± 26.786 ^de^	701.225 ± 61.131 ^de^	561.630 ± 48.962 ^bc^
ethyl phenylethanoate	101-97-3	1259	C_10_H_12_O_2_	ND	489.700 ± 17.656 ^e^	556.300 ± 47.530 ^f^	206.764 ± 4.135 ^b^	322.129 ± 31.726 ^c^	352.771 ± 14.111 ^cd^	413.200 ± 4.132 ^d^
ethyl caprylate	106-32-1	1085	C_10_H_20_O_2_	0.210 ± 0.018 ^a^	38.304 ± 2.762 ^d^	77.526 ± 6.346 g	67.944 ± 4.243 ^f^	12.332 ± 1.176 ^b^	56.771 ± 1.502 ^e^	22.354 ± 0.387 ^c^
ethyl propanoate	105-37-3	686	C_5_H_10_O_2_	ND	517.560 ± 22.560 ^b^	981.120 ± 35.375 ^d^	1226.400 ± 97.342 ^e^	774.213 ± 55.829 ^c^	112.356 ± 3.892 ^a^	580.800 ± 43.849 ^b^
ethyl isobutyrate	97-62-1	728	C_6_H_12_O_2_	ND	ND	55.188 ± 1.104 ^d^	3.552 ± 0.249 ^b^	6.876 ± 0.563 ^c^	ND	ND
ethyl n-valerate	539-82-2	874	C_7_H_14_O_2_	ND	ND	32.280 ± 2.260 ^d^	12.912 ± 0.258 ^b^	ND	20.733 ± 0.415 ^c^	ND
ethyl oleate	111-62-6	2185	C_20_H_38_O_2_	2.451 ± 0.209 ^b^	ND	11.448 ± 0.637 ^c^	ND	ND	50.295 ± 2.012 ^d^	ND
ethyl linoleate	544-35-4	2193	C_20_H_36_O2	ND	ND	8.208 ± 0.701 ^b^	40.776 ± 2.480 ^c^	ND	ND	ND
**Other esters**										
methyl caprate	110-42-9	1282	C_11_H_22_O_2_	ND	384.104 ± 6.653 ^d^	423.658 ± 52.915 ^d^	230.150 ± 22.667 ^bc^	259.881 ± 11.909 ^c^	191.094 ± 6.890 ^b^	237.603 ± 14.453 ^bc^
4-hydroxybutyric acid	591-81-1	1018	C_4_H_7_O_3_	ND	228.427 ± 7.913 ^d^	221.913 ± 11.743 ^cd^	158.713 ± 9.523 ^b^	198.682 ± 7.164 ^c^	205.574 ± 14.390 ^cd^	169.733 ± 11.881 ^b^
txib	6846-50-0	1605	C_16_H_30_O_4_	112.412 ± 8.487 ^a^	1165.722 ± 143.246 ^d^	1178.762 ± 42.501 ^d^	504.371 ± 36.371 ^b^	1165.017 ± 90.991 ^d^	857.818 ± 61.858 ^c^	1207.867 ± 67.251 ^d^
isoamyl decanoate	2306-91-4	1615	C_15_H_30_O_2_	ND	807.121 ± 44.939 ^d^	574.914 ± 15.211 ^b^	811.419 ± 72.120 ^d^	766.740 ± 40.572 ^cd^	ND	684.864 ± 13.697 ^c^
diisobutyl phthalate	84-69-5	1908	C_16_H_22_O_4_	ND	662.163 ± 69.132 ^d^	503.222 ± 5.032 ^c^	446.305 ± 16.092 ^bc^	478.597 ± 36.133 ^bc^	404.190 ± 22.504 ^b^	ND
teksanol	77-68-9	1331	ND	39.739 ± 1.051 ^a^	942.490 ± 18.850 ^e^	221.283 ± 19.668 ^b^	ND	640.200 ± 75.478 ^cd^	551.147 ± 16.534 ^c^	708.996 ± 58.034 ^d^
vinyl acetate	108-05-4	576	ND	ND	20.565 ± 2.167 ^d^	16.672 ± 0.167 ^c^	ND	ND	14.152 ± 1.158 ^bc^	12.396 ± 0.496 ^b^
isobutyl octanoate	5461-06-3	1317	C_12_H_24_O_2_	ND	516.499 ± 17.892 ^e^	485.671 ± 22.256 ^e^	221.345 ± 21.115 ^c^	112.356 ± 9.730 ^b^	324.568 ± 16.865 ^d^	ND
δ-decalactone	705-86-2	1404	C_10_H_18_O_2_	ND	ND	489.300 ± 16.950 ^b^	ND	ND	ND	ND
γ-nonalactone	104-61-0	1284	C_9_H_16_O_2_	ND	16.512 ± 0.858 ^b^	88.244 ± 2.335 ^e^	29.244 ± 1.520 ^c^	60.264 ± 4.346 ^d^	ND	19.704 ± 1.718 ^b^
γ-decalactone	706-14-9	1383	C_10_H_18_O_2_	ND	ND	112.325 ± 7.863 ^b^	ND	ND	ND	ND
γ-caprolactone	695-06-7	986	C_6_H_10_O_2_	ND	ND	22.346 ± 0.387 ^b^	ND	ND	ND	ND
isobutyl propionate	540-42-1	826	C_7_H_14_O_2_	ND	ND	14.880 ± 1.181 ^b^	ND	60.432 ± 3.626 ^c^	ND	ND
isobutyric acid	2445-69-4	955	C_9_H_18_O_2_	ND	ND	13.464 ± 0.539 ^b^	ND	ND	ND	ND
propyl propionate	106-36-5	789	C_6_H_12_O_2_	ND	ND	8.208 ± 0.376 ^b^	ND	ND	ND	ND
isoamyl butylate	106-27-4	1019	C_9_H_18_O_2_	ND	ND	5.088 ± 0.353 ^b^	ND	ND	ND	ND
**Alcohols**										
1-pentanol	71-41-0	761	C_5_H_12_O	ND	205.830 ± 10.292 ^c^	147.275 ± 8.200 ^b^	137.478 ± 10.737 ^b^	141.817 ± 5.113 ^b^	151.472 ± 2.624 ^b^	137.523 ± 5.994 ^b^
1-dodecanol	112-53-8	1457	C_12_H_26_O	0.573 ± 0.035 ^a^	334.224 ± 8.843 ^e^	307.108 ± 16.251 ^e^	191.411 ± 8.343 ^c^	227.642 ± 12.046 ^d^	155.982 ± 9.741 ^b^	208.998 ± 10.450 ^cd^
1-butanol	71-36-3	662	C_4_H_10_O	2.257 ± 0.195 ^a^	430.340 ± 29.815 ^c^	289.039 ± 5.006 ^b^	299.995 ± 21.633 ^b^	294.046 ± 17.886 ^b^	293.252 ± 20.317 ^b^	279.417 ± 15.557 ^b^
isobutyl alcohol	78-83-1	597	C_4_H_10_O	105.581 ± 11.023 ^a^	4433.994 ± 406.382 ^c^	3327.766 ± 295.778 ^b^	3155.172 ± 166.956 ^b^	3187.708 ± 307.411 ^b^	3280.948 ± 56.828 ^b^	2806.563 ± 97.222 ^b^
1-heptanol	111-70-6	960	C_7_H_16_O	32.072 ± 1.156 ^a^	998.475 ± 39.939 ^d^	738.863 ± 33.859 ^b^	728.334 ± 40.552 ^b^	733.172 ± 31.958 ^b^	893.890 ± 35.756 ^c^	659.920 ± 54.017 ^b^
3-octenol	3391-86-4	969	C_8_H_16_O	25.770 ± 1.690 ^a^	542.349 ± 27.117 ^de^	563.245 ± 31.360 ^e^	378.817 ± 36.137 ^b^	420.158 ± 36.629 ^bc^	479.423 ± 37.444 ^cd^	341.961 ± 11.846 ^b^
1-hexanol	111-27-3	860	C_6_H_14_O	19.236 ± 1.763 ^a^	935.539 ± 16.204 ^d^	652.266 ± 59.781 ^c^	619.475 ± 6.195 ^bc^	635.212 ± 16.806 ^bc^	693.327 ± 18.344 ^c^	560.650 ± 49.832 ^b^
3-methylthiopropyl alcohol	505-10-2	912	C_4_H_10_OS	ND	201.238 ± 12.567 ^e^	164.832 ± 13.083 ^d^	118.321 ± 12.798 ^bc^	128.177 ± 5.874 ^c^	122.202 ± 10.583 ^bc^	97.817 ± 2.588 ^b^
benzyl alcohol	100-51-6	1036	C_7_H_8_O	4.170 ± 0.232 ^a^	324.222 ± 8.578 ^e^	356.234 ± 10.687 ^e^	249.895 ± 13.914 ^cd^	262.505 ± 22.428 ^d^	199.869 ± 15.864 ^b^	215.898 ± 16.300 ^bc^
(z)-2,3-butanediol	24347-58-8	743	C_4_H_10_O_2_	101.777 ± 2.036 ^a^	1774.534 ± 93.900 ^b^	2533.951 ± 87.779 ^cd^	2158.838 ± 208.191 ^bc^	2623.290 ± 78.699 ^de^	1811.684 ± 118.800 ^b^	2985.573 ± 273.632 ^e^
hotrienol	29957-43-5	1072	C_10_H_16_O	13.637 ± 0.625 ^a^	392.499 ± 23.875 ^c^	382.457 ± 26.497 ^c^	271.597 ± 24.140 ^b^	284.811 ± 5.696 ^b^	253.654 ± 11.057 ^b^	269.831 ± 28.171 ^b^
3-ethyloctan-3-ol	2051-32-3	1107	C_10_H_22_O	ND	165.414 ± 17.892 ^d^	108.020 ± 1.080 ^bc^	98.871 ± 10.417 ^bc^	103.338 ± 10.789 ^bc^	122.511 ± 6.483 ^c^	96.114 ± 4.404 ^b^
hexanol	104-76-7	995	C_8_H_18_O	0.512 ± 0.040 ^a^	624.766 ± 39.017 ^c^	499.458 ± 4.995 ^b^	477.362 ± 25.260 ^b^	496.366 ± 37.475 ^b^	ND	467.178 ± 30.635 ^b^
cyclooctanol	696-71-9	1147	C_8_H_16_O	8.865 ± 0.320 ^b^	ND	125.682 ± 5.027 ^d^	113.488 ± 0.000 ^c^	ND	134.935 ± 1.349 ^e^	116.765 ± 3.089 ^c^
penten-3-ol	616-25-1	671	C_5_H_10_O	1.206 ± 0.024 ^a^	1375.887 ± 148.825 ^c^	878.405 ± 83.795 ^b^	ND	ND	1030.670 ± 30.920 ^b^	ND
(s)-propylene glycol	4254-15-3	724	C_3_H_8_O_2_	ND	ND	ND	427.920 ± 32.307 ^c^	ND	287.644 ± 25.888 ^b^	654.712 ± 51.966 ^d^
propylene glycol	57-55-6	725	C_3_H_8_O_2_	ND	394.080 ± 14.209 ^b^	520.038 ± 5.200 ^c^	ND	508.569 ± 45.203 ^c^	ND	ND
(s)-3-methyl-1-pentanol	42072-39-9	796	C_6_H_14_O	0.074 ± 0.005 ^a^	ND	71.546 ± 3.279 ^c^	52.434 ± 3.633 ^b^	ND	66.027 ± 5.241 ^c^	ND
2-heptanol	543-49-7	879	C_7_H_16_O	15.703 ± 1.030 ^a^	629.160 ± 43.589 ^d^	552.364 ± 14.614 ^c^	198.720 ± 19.164 ^b^	ND	218.160 ± 12.147 ^b^	ND
(z)-3-hexen-1-ol	928-96-1(正)	868	C_6_H_12_O	ND	96.084 ± 3.328 ^c^	98.832 ± 2.615 ^c^	200.520 ± 19.128 ^e^	ND	60.325 ± 2.413 ^b^	125.100 ± 8.203 ^d^
phenethyl alcohol	60-12-8	1136	C_8_H_10_O	ND	1122.288 ± 73.593 ^c^	248.928 ± 7.468 ^b^	ND	1259.352 ± 90.813 ^cd^	1284.624 ± 96.987 ^d^	ND
1-octanol	111-87-5	1059	C_8_H_18_O	1.011 ± 0.062 ^a^	294.912 ± 17.939 ^d^	74.256 ± 3.403 ^b^	82.488 ± 5.151 ^b^	185.424 ± 12.159 ^c^	303.432 ± 9.103 ^d^	646.200 ± 39.307 ^e^
1-propanol	71-23-8	562	C_3_H_8_O	ND	93.240 ± 9.324 ^c^	185.616 ± 12.993 ^e^	33.288 ± 1.153 ^b^	171.096 ± 2.963 ^e^	349.584 ± 15.238 ^f^	131.352 ± 12.667 ^d^
(r)-2-butanol	14898-79-4	584	ND	ND	ND	8.112 ± 0.354 ^b^	14.760 ± 1.172 ^c^	34.080 ± 2.128 ^e^	ND	21.384 ± 1.697 ^d^
**Acids**										
octanoic acid	124-07-2	1173	C_8_H_16_O_2_	55.346 ± 1.996 ^a^	6868.409 ± 629.500 ^d^	4031.541 ± 358.331 ^bc^	4501.788 ± 400.128 ^c^	4232.770 ± 193.970 ^bc^	3456.568 ± 307.226 ^b^	3477.012 ± 104.310 ^b^
n-decanoic acid	334-48-5	1372	C_10_H_20_O_2_	ND	6043.870 ± 516.388 ^d^	2966.528 ± 207.657 ^c^	2696.471 ± 260.038 ^c^	2417.799 ± 274.609 ^bc^	2262.857 ± 45.257 ^bc^	1848.797 ± 161.174 ^b^
butanoic acid	107-92-6	775	C_4_H_8_O_2_	ND	183.842 ± 3.184 ^c^	134.668 ± 6.733 ^b^	132.010 ± 10.805 ^b^	129.043 ± 9.305 ^b^	139.062 ± 7.358 ^b^	122.402 ± 13.240 ^b^
9-decenoic acid	14436-32-9	1367	C_10_H_18_O_2_	ND	1224.469 ± 76.468 ^d^	808.850 ± 37.066 ^c^	718.849 ± 43.726 ^c^	717.956 ± 31.295 ^c^	799.379 ± 27.691 ^c^	553.493 ± 53.377 ^b^
acetic acid	64-19-7	576	C_2_H_4_O_2_	739.225 ± 29.569 ^a^	7510.545 ± 327.377 ^d^	5260.826 ± 368.258 ^b^	5393.834 ± 494.353 ^b^	6059.155 ± 277.665 ^bc^	6512.075 ± 586.087 ^cd^	5143.817 ± 224.214 ^b^
2-methyl-butanoic acid	116-53-0	817	C_5_H_10_O_2_	155.086 ± 9.685 ^a^	782.741 ± 20.709 ^d^	728.649 ± 66.782 ^cd^	534.598 ± 33.386 ^b^	632.755 ± 38.489 ^bc^	555.536 ± 27.777 ^b^	542.009 ± 21.680 ^b^
heptanoic acid	111-14-8	1073	C_7_H_14_O_2_	ND	485.917 ± 39.774 ^e^	408.038 ± 4.080 ^d^	280.068 ± 19.605 ^c^	394.192 ± 33.680 ^d^	150.860 ± 10.560 ^b^	288.175 ± 15.249 ^c^
2-oxooctanoic acid	328-51-8	1309	C_8_H_14_O_3_	ND	114.051 ± 10.999 ^e^	83.348 ± 5.834 ^bc^	67.059 ± 5.489 ^b^	81.520 ± 5.706 ^bc^	86.314 ± 1.726 ^cd^	100.692 ± 4.389 ^de^
2-methyl- propanoic acid	79-31-2	711	C_4_H_8_O_2_	ND	ND	142.968 ± 11.437 ^c^	ND	686.160 ± 38.204 ^d^	ND	58.260 ± 6.302 ^b^
hexanoic acid	142-62-1	974	C_6_H_12_O_2_	2.789 ± 0.170 ^a^	10.764 ± 1.124 ^a^	ND	361.812 ± 23.726 ^d^	224.880 ± 17.849 ^c^	21.528 ± 0.776 ^ab^	53.052 ± 4.625 ^b^
propanoic acid	79-09-4	676	C_3_H_6_O_2_	ND	ND	51.660 ± 3.725 ^b^	ND	ND	ND	ND
nonanoic acid	112-05-0	1272	C_9_H_18_O2	ND	13.680 ± 1.068 ^bc^	6.864 ± 0.247 ^ab^	44.880 ± 2.244 ^d^	92.640 ± 7.411 ^e^	19.260 ± 1.668 ^c^	37.296 ± 1.345 ^d^
valeric acid	109-52-4	875	C_5_H_10_O2	ND	ND	25.656 ± 0.513 ^b^	ND	ND	ND	ND
**Aldehydes**										
acetaldehyde	75-07-0	408	C_2_H_4_O	114.563 ± 8.948 ^a^	2089.726 ± 165.867 ^c^	1446.705 ± 146.825 ^b^	1418.946 ± 116.146 ^b^	1598.697 ± 162.250 ^b^	1519.862 ± 105.299 ^b^	1335.899 ± 23.138 ^b^
isovaleral	590-86-3	643	C_5_H_10_O	35.654 ± 2.785 ^d^	17.387 ± 0.968 ^a c^	19.246 ± 1.711 ^bc^	13.470 ± 0.486 ^a^	19.648 ± 1.874 ^c^	15.322 ± 0.668 ^ab^	13.159 ± 0.696 ^a^
1-hexanal	66-25-1	806	C_6_H_12_O	88.623 ± 1.772 ^c^	86.230 ± 6.844 ^c^	58.712 ± 1.553 ^b^	35.430 ± 2.455 ^a^	42.741 ± 4.274 ^a^	36.610 ± 3.857 ^a^	42.360 ± 1.527 ^a^
n-octanal	124-13-0	1005	C_8_H_16_O	83.372 ± 3.335 ^c^	59.549 ± 5.955 ^ab^	60.233 ± 2.087 ^ab^	60.536 ± 1.049 ^ab^	53.042 ± 4.005 ^a^	69.814 ± 5.453 ^b^	122.397 ± 6.815 ^d^
nonylaldehyde	124-19-6	1104	C_9_H_18_O	60.392 ± 1.046 ^a^	ND	42.480 ± 2.653 ^a^	64.912 ± 3.246 ^a^	ND	ND	872.538 ± 79.969 ^b^
isobutanal	78-84-2	543	C_4_H_8_O	25.360 ± 2.076 ^d^	ND	7.080 ± 0.511 ^b^	15.816 ± 0.791 ^c^	ND	ND	7.920 ± 0.792 ^b^
**Ketones**										
acetone	67-64-1	455	C_3_H_6_O	ND	13.451 ± 0.485 ^d^	8.852 ± 0.708 ^c^	6.943 ± 0.551 ^b^	7.490 ± 0.225 ^b^	8.234 ± 0.594 ^bc^	7.426 ± 0.324 ^b^
acetoine	513-86-0	717	C_4_H_8_O_2_	86.230 ± 7.664 ^a^	158.089 ± 5.476 ^c^	114.843 ± 6.986 ^b^	122.897 ± 5.632 ^b^	131.471 ± 2.629 ^bc^	157.308 ± 19.330 ^c^	127.983 ± 11.084 ^b^
decan-3-one	928-80-3	1151	C_10_H_20_O	ND	290.986 ± 21.969 ^d^	140.271 ± 2.430 ^b^	128.400 ± 3.852 ^b^	206.693 ± 24.369 ^c^	273.184 ± 19.123 ^d^	ND
biacetyl	431-03-8	691	C_4_H_6_O_2_	4.083 ± 0.071 ^b^	ND	10.112 ± 0.763 ^c^	16.611 ± 1.734 ^d^	16.430 ± 0.999 ^d^	ND	ND
3-octanone	106-68-3	958	C_8_H_16_O	ND	30.912 ± 2.695 ^bc^	24.600 ± 0.852 ^b^	46.080 ± 0.461 ^d^	ND	37.932 ± 2.007 ^c^	76.032 ± 6.758 ^e^
**Ethers**										
dowanol peat	111-35-3	837	C_5_H_12_O_2_	6.007 ± 0.334 ^a^	180.186 ± 15.395 ^de^	104.944 ± 2.777 ^b^	149.404 ± 14.715 ^cd^	198.431 ± 24.384 ^e^	133.791 ± 3.540 ^bc^	120.211 ± 10.819 ^bc^
vinamar	109-92-2	485	C_4_H8O	ND	821.468 ± 42.685 ^e^	571.485 ± 22.859 ^bc^	573.201 ± 29.784 ^bc^	609.315 ± 38.052 ^cd^	671.467 ± 40.844 ^d^	503.150 ± 10.063 ^b^
carbitol	111-90-0	1012	C_6_H_14_O_3_	ND	4.944 ± 0.396 ^c^	1.752 ± 0.109 ^b^	ND	23.736 ± 0.856 ^f^	18.192 ± 0.364 ^e^	12.144 ± 0.421 ^d^

Data are mean values of three independent experiments ± standard deviation. Mean values displaying different letters within each row are significantly different according to the Duncan test at 95% confidence level. pure_F33: fermentation with *S. cerevisiae* F33 in pure. F33_ Hu, F33_ Sc, F33_ Mp, F33_ Pk, F33_ Rl represented the simultaneous mixed fermentation of F33 with *H. uvarum* QTX22, *S. crataegensis* YC30, *M. pulcherrima* YC15, *P. kluyveri* HSP11 and *R. lusitaniae* QTX26, respectively. ND: not detected.

**Table 4 foods-10-01452-t004:** Mean values of OAV for volatile compounds which presented OAV > 1 in Vidal icewine fermented with six fermentation strategies.

Compounds		CAS	RI *	Odor Threshold #	Odor Description	Grape Juice	F33_pure	F33_Hu	F33_Mp	F33_Pk	F33_Sc	F33_Rl	Identification Methods
C13-norisoprenoids													
1	β-damascenone	23726-93-4	1440	0.05 [63]	apple, rose, honey, tobacco, sweet	1414.84 ± 61.67 ^a^	2246.90 ± 178.34 ^b^	3046.07 ± 213.23 ^c^	2973.74 ± 107.22 ^c^	2210.04 ± 192.67 ^b^	3010.71 ± 137.97 ^c^	2506.70 ± 130.25 ^b^	MS, RI, S
Terpenes													
1	naphthalene	91-20-3	1231	60 [64]	pungent, dry, tarry	2.47 ± 0.17 ^a^	11.43 ± 0.79 ^d^	6.95 ± 0.49 ^c^	5.09 ± 0.18 ^b^	5.71 ± 0.60 ^bc^	4.82 ± 0.34 ^b^	4.92 ± 0.22 ^b^	MS, RI, S
2	cinnamene	100-42-5	883	65 [64]	sweet, balsam, floral, plastic	5.91 ± 0.31 ^a^	6.67 ± 0.24 ^ab^	11.57 ± 0.35 ^d^	7.71 ± 0.43 ^bc^	7.95 ± 0.56 ^c^	12.88 ± 0.56 ^e^	7.86 ± 0.61 ^bc^	MS, RI, S
3	linalool	78-70-6	1082	15 [52]	citrus, orange, lemon, floral, waxy	12.17 ± 0.97 ^a^	140.18 ± 5.05 ^bc^	128.16 ± 7.14 ^b^	164.66 ± 6.59 ^cd^	184.74 ± 14.66 ^d^	220.44 ± 17.50 ^e^	150.34 ± 6.55 ^bc^	MS, RI, S
4	citronellol	106-22-9	1179	100 [52]	floral, rose, sweet, green, fruity	ND	3.40 ± 0.38 ^cd^	3.92 ± 0.31 ^de^	2.48 ± 0.07 ^b^	2.97 ± 0.24 ^bc^	4.23 ± 0.11 ^e^	2.50 ± 0.14 ^b^	MS, RI, S
5	d-limonene	5989-27-5	1018	10 [64]	sweet, orange, citrus, terpy	9.99 ± 0.61 ^a^	34.69 ± 2.96 ^c^	33.60 ± 2.87 ^c^	19.83 ± 1.39 ^b^	23.22 ± 1.81 ^b^	19.00 ± 0.83 ^b^	20.90 ± 0.72 ^b^	MS, RI, S
6	α-terpineol	98-55-5	1143	250 [10]	citrus, woody, lemon, nuance	ND	ND	1.39 ± 0.08 ^c^	1.06 ± 0.10 ^b^	ND	ND	ND	MS, RI, S
7	p-xylene	106-42-3	907	58 [65]		ND	ND	ND	3.46 ± 0.07 ^c^	1.81 ± 0.15 ^b^	ND	3.54 ± 0.16 ^c^	MS, RI
8	nerol oxide	1786-08-9	1125	80 [15]	green, vegetative, floral, leafy	2.50 ± 0.25 ^b^	ND	ND	4.34 ± 0.20 ^c^	16.12 ± 1.32 ^e^	6.33 ± 0.60 ^d^	5.25 ± 0.18 ^cd^	MS, RI, S
Acetate esters													
1	isoamyl ethanoate	123-92-2	820	30 [66]	sweet, fruity, banana	1.20 ± 0.07 ^a^	23.47 ± 0.94 ^b^	55.11 ± 0.96 ^d^	41.85 ± 2.51 ^c^	26.31 ± 2.41 ^b^	61.88 ± 6.09 ^d^	26.00 ± 1.80 ^b^	MS, RI, S
2	benzylcarbinyl acetate	103-45-7	1259	250 [66]	sweet, honey, floral, rosy	ND	ND	33.42 ± 2.89 ^c^	25.09 ± 2.71 ^b^	ND	26.21 ± 1.84 ^b^	23.92 ± 1.68 ^b^	MS, RI, S
3	isobutyl acetate	110-19-0	721	1600 [10]	sweet, fruity, banana	ND	ND	1.29 ± 0.10 ^c^	1.04 ± 0.10 ^b^	ND	1.35 ± 0.08 ^c^	1.33 ± 0.17 ^c^	MS, RI, S
4	butyl acetate	123-86-4	774	1800 [52]	sweet, ripe, banana	ND	ND	ND	ND	4.73 ± 0.30 ^d^	1.86 ± 0.05 ^c^	1.18 ± 0.12 ^b^	MS, RI, S
5	propyl acetate	109-60-4	666	4740 [15]	fruity, banana, honey	ND	ND	ND	ND	2.61 ± 0.23 ^c^	1.28 ± 0.11 ^b^	1.16 ± 0.08 ^b^	MS, RI, S
6	hexyl acetate	142-92-7	984	670 [14]	fruity, green, fresh, sweet, banana	ND	ND	ND	1.33 ± 0.14 ^bc^	ND	1.48 ± 0.12 ^c^	1.20 ± 0.10 ^b^	MS, RI, S
Ethyl esters													
1	ethyl dodecylate	106-33-2	1580	500 [10]	waxy, soapy, floral	ND	8.94 ± 0.32 ^b^	19.50 ± 1.35 ^d^	11.48 ± 0.23 ^bc^	12.46 ± 1.25 ^c^	17.73 ± 1.52 ^d^	10.54 ± 0.38 ^bc^	MS, RI, S
2	ethyl heptoate	106-30-9	1099	2.2 [52]	fruity, pineapple, banana	ND	99.01 ± 5.94 ^b^	155.80 ± 6.23 ^c^	97.22 ± 6.07 ^b^	116.67 ± 12.99 ^b^	207.42 ± 12.95 ^d^	98.74 ± 5.50 ^b^	MS, RI, S
3	ethyl caproate	123-66-0	987	8 [52]	sweet, pineapple, fruity, waxy	1.75 ± 0.09 ^a^	555.95 ± 33.82 ^b^	987.03 ± 84.33 ^cd^	870.15 ± 62.75 ^c^	853.94 ± 74.45 ^c^	1108.17 ± 106.87 ^d^	826.09 ± 37.86 ^c^	MS, RI, S
4	ethyl butanoate	105-54-4	785	400 [52]	fruity, sweet, frutti, apple	ND	1.19 ± 0.01 ^b^	2.00 ± 0.04 ^c^	1.27 ± 0.08 ^b^	1.28 ± 0.05 ^b^	2.05 ± 0.02 ^c^	ND	MS, RI, S
5	ethyl dihydrocinnamate	2021-28-5	1359	1.6 [10]	rose, honey, fruity	ND	140.64 ± 16.58 ^b^	401.73 ± 17.51 ^e^	217.08 ± 13.21 ^c^	298.44 ± 7.90 ^d^	488.96 ± 9.78 ^f^	276.94 ± 12.07 ^d^	MS, RI
6	ethyl phenylethanoate	101-97-3	1259	250 [67]	strong, sweet, rosy, honey	ND	1.41 ± 0.08 ^b^	1.96 ± 0.07 ^d^	ND	1.29 ± 0.07 ^b^	2.23 ± 0.06 ^e^	1.65 ± 0.06 ^c^	MS, RI
7	ethyl caprylate	106-32-1	1085	5 [68]	sweet, waxy, fruity, pineapple	ND	11.35 ± 0.41 ^d^	7.66 ± 0.38 ^c^	13.59 ± 1.80 ^de^	2.47 ± 0.19 ^b^	15.51 ± 1.21 ^e^	4.47 ± 0.36 ^b^	MS, RI, S
8	ethyl propanoate	105-37-3	686	550 [52]	fruity, sweet, winey	ND	ND	ND	2.23 ± 0.14 ^e^	1.41 ± 0.13 ^c^	1.78 ± 0.08 ^d^	1.06 ± 0.05 ^b^	MS, RI, S
9	ethyl isobutyrate	97-62-1	728	15 [69]	pungent, etherial, fruity	ND	ND	ND	ND	ND	3.68 ± 0.07 ^b^	ND	MS, RI, S
10	ethyl n-valerate	539-82-2	874	1.5 [12]	fruity, strawberry, sweet, pineapple	ND	13.82 ± 1.46 ^c^	ND	8.61 ± 0.54 ^b^	ND	21.52 ± 1.88 ^d^	ND	MS, RI
Other esters													
1	δ-decalactone	705-86-2	1404	386 [57]	coconut, creamy, fatty, milky	ND	ND	ND	ND	ND	1.27 ± 0.05 ^b^	ND	MS, RI, S
2	γ-nonalactone	104-61-0	1284	30 [57]	coconut, creamy, waxy	ND	ND	ND	ND	2.01 ± 0.16 ^b^	2.94 ± 0.19 ^c^	ND	MS, RI, S
3	γ-decalactone	706-14-9	1383	88 [57]	fruity, creamy, peach	ND	ND	ND	ND	ND	1.28 ± 0.09 ^b^	ND	MS, RI, S
Alcohols													
1	1-pentanol	71-41-0	761	150.2 [64]	fermented, bready	ND	1.01 ± 0.09 ^b^	1.37 ± 0.05 ^c^	ND	ND	ND	ND	MS, RI, S
2	1-heptanol	111-70-6	960	250 [10]	oily, nutty, fatty	ND	3.58 ± 0.38 ^cd^	3.99 ± 0.21 ^d^	2.91 ± 0.15 ^b^	2.93 ± 0.35 ^bc^	2.96 ± 0.17 ^bc^	2.64 ± 0.14 ^b^	MS, RI, S
3	3-octenol	3391-86-4	969	1 [52]	mushroom, earthy, vegetative	25.77 ± 1.95 ^a^	479.42 ± 26.69 ^d^	542.35 ± 24.85 ^e^	378.82 ± 32.37 ^bc^	420.16 ± 16.81 ^cd^	563.25 ± 14.90 ^e^	341.96 ± 21.36 ^b^	MS, RI, S
4	1-hexanol	111-27-3	860	110 [52]	green, fruity, apple-skin, oily	ND	6.30 ± 0.55 ^c^	8.51 ± 0.31 ^d^	5.63 ± 0.49 ^bc^	5.78 ± 0.61 ^bc^	5.93 ± 0.26 ^bc^	5.10 ± 0.27 ^b^	MS, RI, S
5	penten-3-ol	616-25-1	671	400 [52]	whiskey, green, apple	ND	2.58 ± 0.22 ^c^	3.44 ± 0.14 ^d^	ND	ND	2.20 ± 0.14 ^b^	ND	MS, RI, S
6	2-heptanol	543-49-7	879	200 [52]	fruity, green, earthy, bitter	ND	1.09 ± 0.05 ^b^	3.15 ± 0.17 ^c^	ND	ND	2.76 ± 0.35 ^c^	ND	MS, RI, S
7	1-octanol	111-87-5	1059	40 [5]	floral, sweet, rosey, bready	ND	7.59 ± 0.55 ^d^	7.37 ± 0.34 ^d^	2.06 ± 0.09 ^b^	4.64 ± 0.46 ^c^	1.86 ± 0.10 ^b^	16.16 ± 1.17 ^e^	MS, RI, S
Acids													
1	octanoic acid	124-07-2	1173	500 [66]	rancid, soapy, cheesy, fatty	ND	6.91 ± 0.21 ^b^	13.74 ± 0.63 ^e^	9.00 ± 0.72 ^d^	8.47 ± 0.52 ^cd^	8.06 ± 0.74 ^bd^	6.95 ± 0.61 ^bc^	MS, RI, S
2	n-decanoic acid	334-48-5	1372	1000 [68]	soapy, waxy, fruity	ND	2.26 ± 0.12 ^bc^	6.04 ± 0.52 ^e^	2.70 ± 0.10 ^cd^	2.42 ± 0.23 ^bd^	2.97 ± 0.19 ^d^	1.85 ± 0.10 ^b^	MS, RI, S
3	9-decenoic acid	14436-32-9	1367	40 [52]	waxy, creamy, fatty	ND	19.98 ± 0.80 ^c^	30.61 ± 2.92 ^d^	17.97 ± 2.00 ^bc^	17.95 ± 0.72 ^bc^	20.22 ± 0.54 ^c^	13.84 ± 1.27 ^b^	MS, RI, S
4	2-methyl-butanoic acid	116-53-0	817	33 [70]	fruity	4.70 ± 0.25 ^a^	16.83 ± 0.17 ^bc^	23.72 ± 0.63 ^d^	16.20 ± 1.60 ^b^	19.17 ± 1.17 ^c^	22.08 ± 0.80 ^d^	16.43 ± 1.08 ^b^	MS, RI
5	2-methyl- propanoic acid	79-31-2	711	230 [68]	acidic, sour, cheesy	ND	ND	ND	ND	2.98 ± 0.21 ^b^	ND	ND	MS, RI
Aldehydes													
1	acetaldehyde	75-07-0	408	186 [64]	pungent, fresh	ND	8.17 ± 0.78 ^b^	11.24 ± 0.88 ^c^	7.63 ± 0.20 ^b^	8.60 ± 0.34 ^b^	7.78 ± 0.54 ^b^	7.18 ± 0.88 ^b^	MS, RI, S
2	isovaleral	590-86-3	643	0.35 [52]	fruity, dry, green, chocolate	101.87 ± 7.35 ^c^	43.78 ± 4.22 ^ab^	49.68 ± 4.33 ^ab^	38.49 ± 3.08 ^a^	56.14 ± 6.47 ^b^	54.99 ± 5.25 ^b^	37.60 ± 3.28 ^a^	MS, RI, S
3	1-hexanal	66-25-1	806	4.5 [52]	green, woody, vegetative	19.69 ± 0.52 ^c^	8.14 ± 0.67 ^a^	19.16 ± 0.96 ^c^	7.87 ± 0.47 ^a^	9.50 ± 0.19 ^a^	13.05 ± 1.14 ^b^	9.41 ± 0.82 ^a^	MS, RI, S
4	n-octanal	124-13-0	1005	15 [52]	aldehyde, green	5.56 ± 0.31 ^b^	4.65 ± 0.23 ^ab^	3.97 ± 0.12 ^a^	4.04 ± 0.50 ^a^	3.54 ± 0.18 ^a^	4.02 ± 0.20 ^a^	8.16 ± 0.83 ^c^	MS, RI, S
5	nonylaldehyde	124-19-6	1104	15 [52]	cucumber, melon, rindy	4.03 ± 0.34 ^b^	ND	ND	4.33 ± 0.35 ^b^	ND	2.83 ± 0.10 ^ab^	58.17 ± 3.63 ^c^	MS, RI
6	isobutanal	78-84-2	543	6 [71]	fresh, herbal, green	4.23 ± 0.40 ^d^	ND	ND	2.64 ± 0.07 ^c^	ND	1.18 ± 0.04 ^b^	1.32 ± 0.04 ^b^	MS, RI, S
Ketones													
1	3-octanone	106-68-3	958	21.4 [52]	mushroom, ketonic, cheesy	ND	1.77 ± 0.19 ^cd^	1.44 ± 0.01 ^bc^	2.15 ± 0.21 ^d^	ND	1.15 ± 0.11 ^b^	3.55 ± 0.34 ^e^	MS, RI, S
Ethers													
1	dowanol peat	111-35-3	837	100 [52]	fruit	ND	1.34 ± 0.07 ^cd^	1.80 ± 0.08 ^e^	1.49 ± 0.08 ^d^	1.98 ± 0.16 ^e^	1.05 ± 0.04 ^b^	1.20 ± 0.12 ^bc^	MS, RI

Data are mean values of three independent experiments ± standard deviation. Mean values displaying different letters within each row are significantly different according to the Duncan test at 95% confidence level. pure_F33: fermentation with *S. cerevisiae* F33 in pure. F33_ Hu, F33_ Sc, F33_ Mp, F33_ Pk, F33_ Rl represented the simultaneous mixed fermentation of F33 with *H. uvarum* QTX22, *S. crataegensis* YC30, *M. pulcherrima* YC15, *P. kluyveri* HSP11 and *R. lusitaniae* QTX26, respectively. ND: not detected. Odor description are from flavornet database (http://www.flavornet.org; http://www.thegoodscentscompany.com, accessed on 11 September 2019). * RI: retention index on a DB-WAX column. Identification methods: MS, identified by the MS spectra; RI, identified by comparison with retention indices of standards; S, identified by comparison to standards. # Odor thresholds are referred from literatures.

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
