# Peer review of "Effects of Simultaneous Co-Fermentation of Five Indigenous Non-*Saccharomyces* Strains with *S. cerevisiae* on Vidal Icewine Aroma Quality"

_foods, 2021, doi:10.3390/foods10071452_

Round 1

Reviewer 1 Report

Dear Author:

The manuscript entitled “Simultaneous fermentation of S. crataegensis YC30 and S. cerevisiae increased esters aroma compounds lead to wine fruity and floral aroma enhancement” posses in my opinion novelty novelty related to the use of the yeasts S. crataegensis and R. lusitaniae. Additionally, specific studies on Ice wine are less reported in scientific literature than regular wines. The materials and methods are appropriated and studied a huge number of phenolic and aroma compounds. Nevertheless, I propose some corrections/suggestions that in my opinion could improve the manuscript.

References => references in all the manuscript are “……. et al., XXX”. The manuscrpits that I have lately read from Foods journal usually use the reference in brackets in the text “[X]” and later the numbers are referenced to the proper citation in the Reference section. Check if the citations are according to the journal indications or they must be changed.

Introduction => The introduction expose properly the icewine and general non-Saccharomyces. However, in some occasions (paragraph 3 of introduction) deals with specific non-Saccharomyces (P. fermentans, Torualaspora delbrueckii, Zygosaccharomyces bailii or Starmerella bacillaris) that are not studied in the manuscript while other ones used in the study are not mentioned in the introduction. R. lusitaniae or S. cratagensis can be difficult to report in previous studies but H. uvarum, P. kluyveri or M. pulcherrima possess extended specific literature and specific reviews that make it easy to expose the main idea about using them in winemaking in one line in the introduction.

Martin, V., Valera, M. J., Medina, K., Boido, E., & Carrau, F. (2018). Oenological impact of the Hanseniaspora/Kloeckera yeast genus on wines—a review. Fermentation4(3), 76.

Vicente, J., Calderón, F., Santos, A., Marquina, D., & Benito, S. (2021). High potential of Pichia kluyveri and other Pichia species in wine technology. International Journal of Molecular Sciences22(3), 1196.

Additionally, in some sections of the manuscript some results are compared to non-Saccaharomyces that are not used in the study (Example: “…. have previously reported that the mixed fermentation of Torulaspora delbrueckii and S. cerevisiae at a ratio of 1:1 can inhibit the bio-mass of S. cerevisiae and control the entire fermentation process t...”). I think that the manuscript would improve if it were possible to compare the results with specific studies related to the studied species (Although in some occasion could be no available studies that justify to use other non-Saccharomyces no used in the study).

Example of specific studies: Section 3.4.1 Varietal and aroma substances => “P. kluyveri HSP14 significantly improved the content of terpenes sub-stances,”

Previous specific studies on P. kluyveri observed this phenomenon that could be related to a higher β-glucosidase activity. So it is possible to contrast those results with specific ones of P. kluyveri.

Whitener, M.B.; Stanstrup, J.; Carlin, S.; Divol, B.; Du Toit, M.; Vrhovsek, U. Effect of non-Saccharomyces yeasts on the volatile chemical profile of Shiraz wine. Aust. J. Grape Wine Res. 2017, 23, 179–192.

Escribano, R.; González-Arenzana, L.; Garijo, P.; Berlanas, C.; López-Alfaro, I.; López, R.; Gutiérrez, A.R.; Santamaría, P. Screening of enzymatic activities within different enological non-Saccharomyces yeasts. J. Food Sci. Technol. 2017, 54, 1555–1564.

Example section 3.2. Effect of fermentation strategies on physicochemical characteristics in Vidal icewine => “Pk strain produced the highest levels of pyruvate, malic, succinic and acetic acid (0.564 g/L),”. A previous study reports a very high production of succinic acid involving sequential fermentations with P. kluyveri. So data can be compared

Lu, Y.; Voon, M.K.W.; Chua, J.-Y.; Huang, D.; Lee, P.-R.; Liu, S.-Q. The effects of co-and sequential inoculation of Torulaspora delbrueckii and Pichia kluyveri on chemical compositions of durian wine. Appl. Microbiol. Biotechnol. 2017, 101, 7853–7863.

Best Regards.

Author Response

Foods

Title: Simultaneous fermentation of S. crataegensis YC30 and S. cerevisiae increased esters aroma compounds lead to wine fruity and floral aroma enhancement

Dear Editor,

Thank you very much for your attention. We very much appreciate your patience and positive constructive comments on our manuscript (FOODS-1217095). The affirmation of our work has greatly encouraged us. According to those helpful comments, we have made a careful revision on the manuscript. All revisions are explained as follows:

All changes are displayed in red font in revised manuscript.

Reviewer 2 Report

Article ID: Foods_1217095

Title: Simultaneous fermentation of S. crataegensis YC30 and S. cerevisiae increased esters aroma compounds lead to wine fruity and floral aroma enhancement

Authors: Qian Ge, Chunfeng Guo, Jing Zhang, Yue Yan, Zhao Danqing, Caihong Li, Xiangyu Sun, Tingting Ma, Tianli Yue, Yahong Yuan

The manuscript presents the characteristics of 6 alcoholic beverages produced from Vidal grape-must fermented with single or mixed culture of Saccharomyces cerevisiae F33 with five indigenous non-Saccharomyces yeast strains (Hanseniaspora uvarum QTX22, Saccharomycopsis crataegensis YC30, Pichia kluyveri HSP14, Metschnikowia pulcherrima YC12, Rhodosporidiobolus lusitaniae QTX15). General physicochemical analysis, including alcohol content, titratable acidity, sugars, organic acids, phenolic, volatile compounds and sensory analysis were performed, prior and after fermentation. The authors concluded that simultaneous fermentation increased the concentrations of total phenolic compounds and total volatile compounds especially in mixed-culture fermentation with S. crataegensis YC30. The subject is not entirely new since numerous studies have shown that selected non-Saccharomyces yeast strains used as adjuncts of S. cerevisiae result in wines judged to be of better quality than those produced by the S. cerevisiae only.

In this line, I would like to see citations to some of the pioneering work in this area and only then the more recent work, which may provide additional insights into the subject.

I have some concern about the experimental design, since the number of biological replicates was never explicitly mentioned in the manuscript. How many fermentations were carried out in triplicate? The word “duplicate” appears only once in subheading “2.7. Sensory analysis” referring to the wine samples which were identified in duplicate.

Points need review:

Title: the title does not completely reflect the content of the work. Make it shorter;

The nomenclature of the species should be in italic. Check all the text, including the references;

Material and Methods:

The methodologies should be described to an extent that makes understandable about what has been really done, and why these procedures were conducted.  There are aspects that could be clarified in the manuscript.  For example, the yeast assimilable nitrogen (YAN) was not measured in grape-juice nor in the wines despite of being one parameter that strongly affect final wine composition, in particular in volatile compounds released by yeasts, whose YAN requirements are so variable. Also, the values of volatile acidity found in the wines fermented with different non-Saccharomyces are not mentioned in the text, despite of being a crucial point on the selection of the best strain to be used in the industry;

In “2.1. Yeast strains” Add a short characterization of the yeast S. crataegensis YC30;

In “2.3. Grape juice” please indicate the methodologies used for the determination of the following parameters: ethanol, sugar, TA;

Other aspects related to conducting and monitoring fermentations must be included, namely the evolution of sugar breakdown, ethanol formation along fermentations, and how the completion of fermentations were determined.  In “2.4: fermentation strategies” the authors referred the article of Wei et al 2020 saying the fermentation strategies are similar, but with modifications. Please rewrite the procedure briefly and concisely so makes possible repeating your conditions.;

The concentration of yeast cells added in each assay, for each of the yeast strain tested, must be included;

In respect to “2.5. Assay for organic acids and polyphenol compounds” is recommended that authors give separated information about the operating conditions for determination of the organic acids or for determination of phenolic compounds. It is acceptable that well-known methods can be briefly described, and a reference should be given for more detailed information. In this subheading 2.5 the methodology used must be based on results already validated and published in the bibliography;

Volatile aroma compounds were determined by SPME-GC-MS. It is not clear if the major volatile compounds (acetaldehyde, ethyl acetate, methanol, 1-propanol, 2-methyl-1-propanol, 2-methyl-1-butanol, 3-methyl-1-butanol and 2-phenylethanol) and the minor volatile compounds (e.g. esters) were identified by the same methodology;

Results and discussion

Results must be clear and concise avoiding overlapping between figures illustration and the text. I suggest it must be focused on the topic, in which authors can provide what makes this work original/innovative when compared to others. I strongly recommend a careful revision of all the section.

“3.1. Fermentation performance of yeasts

 The methodologies used for determination of ” the parameters fermentation rate, yeasts biomass, cell numbers are not described in material and Methods section;

Clarify the concept and rephrase the statement “Figure 1 A shows the growth kinetic characteristics of …… with five non-Saccharomyces yeasts” since Figure 1 A shows the amount of CO2 released during grape-must fermentation …..;

In table 1- there is a mistake on the % of ethanol in grape-juice (10.12) and F33-pure (ND)

From the figure 1A is not clear that simultaneous fermentation decreased and slowed down the fermentation rate of wine that the maximum;

Figure 1 A. is practically unreadable. Moreover, Figure 1 caption must be improved with a description of the illustration, because it includes fermentation kinetics and evolution of the number of CFUs/ml along the fermentation;

All the figures need improvement since they must be comprehensible without being necessary reading the main text. In respect of the composite Figures 2, 3 and 4 some of the panels need to be enlarged in order for all the elements to be visible;

A correlation between volatile substances (namely those present above the perception threshold) and sensory analysis should be carried out.

Author Response

(The authors gave the same response as above.)

Reviewer 3 Report

The article reports a large diversity of results deriving from fementation trials of a icewine. However, the results are at most preliminary because of the experimental design and so are not enough to validate the claims of the authors.

The main limitations are:

1. Experiments are microvinifications with 150 mL, so can only be considered as preliminary. We do not understand how this volume is enough for sensory analysis by 18 tasters.

2. Table 1, grape juice has ~72 g/L of sugar and 10% ethanol?

3. In complex products like wine, OAV values are only indicative of odour qualities and may not reflect wine flavour.

4. Different aroma descriptors in spider plots are not proof of different wine aroma. Authors must perform tests to determine overall wine differences (e.g. triangular tests). This drawback is common in published papers and authors should be refrained from assuming that different aromatic profiles are reflected in the ability to distinguish wines.

5. Tasting was only based on orthonasal aroma which is not enough to wine characterisation.

Besides, all text should be reviewed with many grammar and notation mistakes, including bibliographic references.

Author Response

(The authors gave the same response as above.)

Round 2

Reviewer 2 Report

Line 113-114: microvessel capped with sulfuric acid . Clarify the procedure/condition used

Lines 125-127: One milliliter of fermenting grape juice was 126 collected periodically (Day 0, 1, 2, 3, 4, 5, 7, 9, 12 and 13) and plated on WL nutrient agar to facilitate yeast population counts. Clarify the procedure. If you direct spread 1mL of fermenting juice on the surface of nutrient agar in a Petri dish, there will be too many colonies on the plate that it will be impossible to actually count the number of CFUs.  Appropriate dilution is required;

Figures still need improvement since in composite figures 2,3 and 4 some of the panels need to be enlarged to make them readable.

English editing is required with assistance of a native English speaker

Author Response

Foods

Title: Effects of simultaneous co-fermentation of five indigenous non-Saccharomyces strains with S. cerevisiae on Vidal icewine aroma quality

Dear Editor,

Thank you very much for your attention. We very much appreciate your patience and positive constructive comments on our manuscript (FOODS-1217095). The affirmation of our work has greatly encouraged us. According to those helpful comments, we have made a careful revision on the manuscript. All revisions are explained as follows:

All changes are displayed in red font in revised manuscript.

Reviewer 3 Report

The paper has been adequately corrected.

Author Response

Thank you very much for your attention. We very much appreciate your patience and positive constructive comments on our manuscript